

# Characteristics of Interannual Variability in Space-based XCO₂ Global Observations

Yifan Guan[1], Gretchen Keppel-Aleks[1], Scott C. Doney[2], Christof Petri[3], Dave Pollard[4], Debra Wunch[5], Frank Hase[6], Hirofumi Ohyama[7], Isamu Morino[7], Justus Notholt[3], Kei Shiomi[9], Kim Strong[5], Rigel Kivi[10], Matthias Buschmann[3], Nicholas Deutscher[12], Paul Wennberg[11], Ralf Sussmann[8], Voltaire A. Velazco[12,13], Yao Té[14]

[1]Department of Climate and Space Sciences and Engineering, University of Michigan, Ann Arbor, MI, 48105, USA
[2]Department of Environmental Sciences, University of Virginia, Charlottesville, VA, 22903, USA
[3]Institute of Environmental Physics, Physics Department, University of Bremen, Bremen, 28359, Germany
[4]National Institute of Water and Atmospheric Research, Omakau, 9377, New Zealand
[5]Department of Physics, University of Toronto, Toronto, M5S 1A7, Canada
[6]Institute of Meteorology and Climate Research - Atmospheric Trace Gases and Remote Sensing, Karlsruhe Institute of Technology, Eggenstein-Leopoldshafen, 76344, Germany
[7]National Institute for Environmental Studies, Tsukuba, Ibaraki 305-0053, Japan
[8]Institute of Meteorology and Climate Research - Atmospheric Environmental Research, Karlsruhe Institute of Technology, Garmisch-Partenkirchen, 82467, Germany
[9]Earth Observation Research Center, Japan Aerospace Exploration Agency, Tsukuba, Ibaraki, 305-8505, Japan
[10]Space and Earth Observation Centre, Finnish Meteorological Institute, Sodankylä, 99600, Finland
[11]Divisions of Engineering and Applied Science and Geological and Planetary Science, California Institute of Technology, Pasadena, CA, 91125, USA
[12]Centre for Atmospheric Chemistry, School of Earth, Atmospheric and Life Sciences, University of Wollongong, Wollongong, 2522, Australia
[13]Deutscher Wetterdienst, Meteorological Observatory, Hohenpeissenberg, 82383, Germany
[14]Laboratoire d'Etudes du Rayonnement et de la Matière en Astrophysique et Atmosphères, French National Centre for Scientific Research, Paris, 75016, France

*Correspondence to*: Yifan Guan (guanyf@umich.edu)

**Abstract.** Atmospheric carbon dioxide (CO₂) accounts for the largest radiative forcing among anthropogenic greenhouse gases. There is, therefore, a pressing need to understand the rate at which CO₂ accumulates in the atmosphere, including the interannual variations (IAV) in this rate. IAV in the CO₂ growth rate is a small signal relative to the long-term trend and the mean annual cycle of atmospheric CO₂, and IAV is tied to climatic variations that may provide insights into long-term carbon-climate feedbacks. Observations from the Orbiting Carbon Observatory-2 (OCO-2) mission offer a new opportunity to refine our understanding of atmospheric CO₂ IAV since the satellite can measure over remote terrestrial regions and the open ocean where traditional in situ CO₂ monitoring is difficult. In this study, we analyze the IAV of column-averaged dry air CO₂ mole fraction (XCO₂) from OCO-2 between September 2014 to June 2021. The amplitude of IAV variations is up to 1.2 ppm over the continents and around 0.4 ppm over the open ocean. Across all latitudes, the OCO-2 detected XCO₂ IAV shows a clear relationship with ENSO-driven variations that originate in the tropics and are transported poleward. The XCO₂ IAV timeseries shows similar zonal patterns compared to ground-based in situ observations and with column observations from the Total Carbon Column Observing Network (TCCON). At lower degrees of aggregation (i.e., 5°x5° grid cells), there are larger inconsistencies with TCCON suggesting that one or both of the observing systems are affected by bias or





systematic retrieval issues that are of a similar magnitude to the IAV signal. Our results suggest that OCO-2 IAV provides meaningful information about climate-driven variations in carbon fluxes and provides new opportunities to monitor climate-driven variations in $CO_2$ over open ocean and remote regions.

## 1 Introduction

Increasing atmospheric $CO_2$ concentration from anthropogenic emissions is the major driver of the observed warming of Earth's climate since the industrial revolution (IPCC, 2021). Although $CO_2$ accumulation in the atmosphere generally is ~45% of anthropogenic emissions on a multi-year average (Ciais et al., 2013; Friedlingstein et al., 2019),

the growth rate shows substantial interannual variability (Conway et al., 1994). The difference between emissions and the atmospheric $CO_2$ growth rate results from net $CO_2$ uptake by oceans and terrestrial ecosystems (Prentice et al., 2001; Doney et al., 2009), and the fluctuations reflect variations in the strength of those sinks due to climate variations (Peters et al., 2017; Friedlingstein et al., 2019). Much research has suggested that interannual variability (IAV) in the growth rate is predominantly due to variations in terrestrial ecosystem carbon uptake (Marcolla et al., 2017), even

though the average uptake is roughly comparable between land and ocean (Le Quéré et al., 2009). Existing atmospheric $CO_2$ observations from surface flask sampling and in situ networks have been used to estimate global- and regional-scale interannual variability in $CO_2$ fluxes (Gurney et al., 2003; Peylin et al., 2013; Keppel-Aleks et al., 2014; Piao et al., 2020). We note, however, that the surface observing network is located primarily on land and coastal sites, and more subtle ocean flux signals may be obscured by the large IAV in terrestrial fluxes.

Previous analyses of surface $CO_2$ IAV has shown a strong relationship with the phase and intensity of El Niño–Southern Oscillation (ENSO) (Le Quéré et al., 2009; Schwalm et al., 2011). ENSO variations originate from coupled ocean-atmosphere dynamics that are reflected in large wind and sea surface temperature anomalies over the central and eastern Pacific Ocean. ENSO affects the climate of much of the tropics and subtropics via atmospheric

teleconnections on timescales of 2-7 years (Timmermann et al., 2018). On land, suppressed precipitation and high temperature associated with positive phases of ENSO (El Niño conditions) suppress $CO_2$ uptake by tropical ecosystems, while promoting fires that further reduce the $CO_2$ uptake by lands (Feely et al., 2002; McKinley et al., 2004; Piao et al., 2009; Wang et al., 2013). Although of smaller magnitude, the equatorial Pacific Ocean experiences weakening of the easterly trade winds and suppression of ventilation of deep, cold, carbon-rich waters to the surface

during an El Nino, reducing the efflux of natural $CO_2$ to the atmosphere (Patra et al., 2005 ; Chatterjee et al., 2017).

Chatterjee et al., (2017) were able to directly observe the ocean flux-driven signal on atmospheric $CO_2$ from El Nino for the first-time using $XCO_2$ (column-averaged dry air $CO_2$ mole fraction) observed over the ocean by NASA's OCO-2 satellite. Space-based observations from OCO-2, which launched in July 2014, provide novel opportunities to

characterize the patterns of IAV in $XCO_2$ in areas that were previously not directly observed by existing monitoring networks. The IAV in $XCO_2$ is being used implicitly for flux attribution in inverse modeling studies. These exciting



results, however, must be tempered by an awareness that atmospheric $CO_2$ IAV is a relatively small signal. For example, IAV in the surface network is about 1 ppm in scale compared to a seasonal amplitude of around 10 ppm in northern high latitudes. OCO-2 measures column averaged $CO_2$, so its measurements are sensitive to variations in the boundary layer mole fraction, which is in direct contact with the land or atmospheric fluxes, but also variations in the free troposphere and stratosphere, where flux signals are generally smaller than those observed at the surface (Olsen and Randerson, 2004). Furthermore, variations in the free troposphere are expected to have relatively long correlation length scales due to efficient mixing, making it important to consider the spatial scales at which $XCO_2$ observations provide unique information. This is especially important in light of analysis which suggests that the error variance budget in OCO-2 observations is large and contains substantial spatially coherent signal (Torres et al., 2019; Mitchell et al., submitted).

In this paper, we analyze $XCO_2$ from OCO-2 to characterize spatiotemporal patterns in IAV at near-global scale, over both land and ocean, and relate $XCO_2$ variations to ENSO conditions. We contextualize the information contained in OCO-2 observations by comparing with ground-based TCCON $XCO_2$ and with surface measurements of $CO_2$. Finally, we use these comparisons to emphasize the spatial scales at which the IAV signal emerges from instrumental noise.

## 2  Data and Methods

### 2.1 Datasets

#### 2.1.1 OCO-2 observatory

We analyzed IAV in dry air, column-average mole fraction $XCO_2$ inferred from OCO-2 satellite observations. The OCO-2 observatory was launched in July 2014 and has measured passive, reflected solar near infrared $CO_2$ and $O_2$ absorption spectra using grating spectrometers since September 2014 (Eldering et al., 2017). $XCO_2$ data are retrieved from the measured spectra using the Atmospheric $CO_2$ Observations from Space (ACOS) optimal estimation algorithm, which is a full physics algorithm that takes into account $XCO_2$ and other physical parameters, including surface pressure, surface albedo, temperature, and water vapor profile in its state vector (Crisp et al., 2010; O'Dell et al., 2018). The satellite flies in a polar and sun-synchronous orbit that repeats every 16 days, with three different observing modes of OCO-2, namely nadir (land only, views the ground directly below the spacecraft), glint (over ocean and land, views just off the peak of the specularly reflected sunlight), and target (typically for comparison with specific ground-based or airborne measurements) (Crisp et al., 2012; Crisp et al., 2017). We use the version 10 OCO-2 Level 2 bias-corrected $XCO_2$ data product from Goddard Earth Sciences Data and Information Services Center (GES DISC) Archive: https://disc.gsfc.nasa.gov/datasets/OCO2_L2_Lite_FP_10r/summary), which has been validated with collocated ground-based measurements from the Total Carbon Column Observing Network (TCCON; discussed in more detail in Section 2.2). After filtering and bias correction, the OCO-2 $XCO_2$ retrievals agree well with TCCON in nadir, glint, and target observation modes, and generally have absolute median differences less than 0.4 ppm and Root Mean Square differences less than 1.5 ppm (O'Dell et al., 2012; Wunch et al., 2017).



### 2.1.2 TCCON

We corroborate patterns of $XCO_2$ IAV from OCO-2 with those from TCCON, a ground-based network of Fourier transform spectrometers that measure direct solar absorption spectra in the near infrared (Wunch et al., 2011).
Retrievals of $XCO_2$ and other gases are computed using the GGG algorithm, a nonlinear least-squares spectral fitting algorithm. The TCCON retrievals are tied to the World Meteorological Organization (WMO) X2007 $CO_2$ scale via calibration with aircraft and AirCore profiles above the TCCON sites (Karion et al., 2010; Wunch et al., 2010). This ensures an accuracy and precision of ~0.6 ppm(1sigma) throughout the network (Washenfelder et al., 2006; Messerschmidt et al., 2010; Deutscher et al., 2010, Wunch et al., 2010). TCCON has been used widely as a validation
standard by providing independent measurements to compare with multiple satellite $XCO_2$ retrievals including OCO-2. In previous work Sussmann and Rettinger (2020) have demonstrated a concept to retrieve annual growth rates of $XCO_2$ from TCCON data, which are regionally to hemispherically representative in spite of the non-uniform sampling in time and space inherent to the ground-based network. In our study, we focus on IAV in the $XCO_2$ timeseries from 26 TCCON sites (Table 1, Fig.1) that have at least 3 years of observational coverage within the period from September
2014 to June 2021. These TCCON data have been filtered using the standard filter that is based on a measure of cloudiness and limits the solar zenith angle. Data are publicly available from the TCCON Data Archive (https://tccondata.org/) hosted by the California Institute of Technology.

**Table 1. TCCON Column-Averaged Dry-Air Mole Fractions of CO₂ (GGG2014 Data)**


| Region | Site | Acronym | Latitude | Longitude | Start Date | End Date | Publication |
|---|---|---|---|---|---|---|---|
| **Polar Northern Hemisphere (60-90°N)** | Eureka (NU) | eu | 80.05 | -86.42 | 2010-07 | 2020-07 | Strong, K. et al., 2017 |
| | Ny Ålesund | sp | 78.90 | 11.90 | 2014-04 | 2019-09 | Notholt, J. et al., 2019 |
| | Sodankylä (FI) | so | 67.37 | 26.63 | 2009-05 | 2020-10 | Kivi, R. et al., 2017 |
| **Temperate Northern Hemisphere (20-60°N)** | East Trout Lake(SK) | et | 54.35 | -104.99 | 2016-10 | 2020-09 | Wunch, D., et al., 2017 |
| | Bialystok (PL) | bi | 53.23 | 23.03 | 2009-03 | 2018-10 | Deutscher, N. et al., 2017 |
| | Bremen (DE) | br | 53.10 | 8.85 | 2010-01 | 2020-06 | Notholt, J. et al., 2017 |
| | Karlsruhe (DE) | ka | 49.10 | 8.44 | 2010-04 | 2020-11 | Hase, F. et al., 2017 |
| | Paris (FR) | pr | 48.97 | 2.37 | 2014-09 | 2020-06 | Te, Y. et al., 2017 |
| | Orléans (FR) | or | 47.97 | 2.11 | 2009-08 | 2020-06 | Warneke, T. et al., 2017 |
| | Garmisch (DE) | gm | 47.48 | 11.06 | 2007-07 | 2020-06 | Sussmann, R. et al., 2017 |
| | Zugspitze (DE) | zs | 47.42 | 10.98 | 2015-04 | 2020-06 | Sussmann, R. et al., 2018 |
| | Park Falls (US) | pa | 45.95 | -90.27 | 2004-06 | 2020-12 | Wennberg, P. O. et al., 2017 |
| | Rikubetsu (JP) | rj | 43.46 | 143.77 | 2013-11 | 2019-09 | Morino, I. et al., 2017 |
| | Lamont (US) | oc | 36.60 | -97.49 | 2008-07 | 2020-12 | Wennberg, P. O. et al., 2017 |
| | Anmyeondo (KR) | an | 36.58 | 126.33 | 2015-02 | 2018-04 | Goo, T.-Y. et al., 2017 |
| | Tsukuba (JP) | tk | 36.05 | 140.12 | 2011-08 | 2019-09 | Morino, I. et al., 2017 |
| | Edwards (US) | df | 34.96 | -117.88 | 2013-07 | 2020-12 | Iraci, L. et al., 2017 |
| | Caltech (US) | ci | 34.14 | -118.13 | 2012-09 | 2020-12 | Wennberg, P. O. et al., 2017 |
| | Saga (JP) | js | 33.24 | 130.29 | 2011-07 | 2020-12 | Shiomi, K.et al., 2017 |
| | Izana (ES) | iz | 28.30 | -16.50 | 2007-05 | 2021-02 | Blumenstock, T. et al., 2017 |



| Tropical Northern Hemisphere (0-20°N) | Burgos (PH) | bu | 18.53 | 120.65 | 2017-03 | 2020-03 | Morino, I., et al., 2018 |
|---|---|---|---|---|---|---|---|
| Tropical Southern Hemisphere (0-20°S) | Ascension Island (SH) | ae | -7.92 | -14.33 | 2012-05 | 2018-10 | Feist, D. G. et al., 2017 |
| | Darwin (AU) | db | -12.46 | 130.94 | 2005-08 | 2020-04 | Griffith, D. W. T., et al., 2017 |
| Temperate Southern Hemisphere (20-60°S) | Réunion Island (RE) | ra | -20.90 | 55.49 | 2011-09 | 2020-07 | De Maziere, M. et al., 2017 |
| | Wollongong (AU) | wg | -34.41 | 150.88 | 2008-06 | 2020-06 | Griffith, D. W. T. et al., 2017 |
| | Lauder (NZ) | ll | -45.04 | 169.68 | 2010-02 | 2018-10 | Sherlock, V. et al., 2017 |

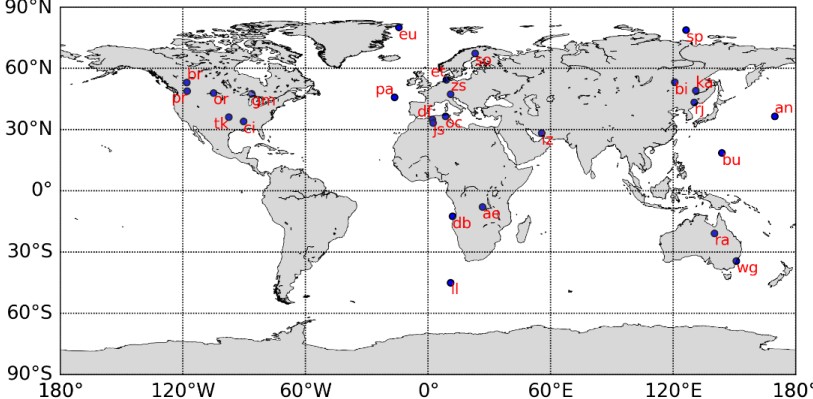

Figure 1. Map showing the locations and the acronyms of the TCCON sites.

### 2.1.3 Marine Boundary Layer Observations

To explore differences in surface and column-average $CO_2$ IAV, we analyze IAV in the surface $CO_2$ mole fraction at marine boundary layer (MBL) sites in the NOAA (National Oceanic and Atmospheric Administration) cooperative sampling network. At these sites, boundary layer $CO_2$ is measured using weekly flask samples (Masarie and Tans, 1995; Dlugokencky et al., 2016). MBL sites are typically far away from anthropogenic sources and regions of active terrestrial exchange, so they provide an estimate for large-scale patterns in the global background $CO_2$ concentration. The surface MBL dry air mole fraction data has an accuracy level of about 0.1 ppm. In this study, we select 16 sites with at least 80% data coverage for the approximately 7-year period overlapping with OCO-2 (Table 2, Fig.2), and the data are aggregated into four north-south zones for comparison with OCO-2 $XCO_2$: northern and southern hemisphere tropical (0 - 20°) and Northern Hemisphere/Southern Hemisphere extratropical zones (20-60°). Each belt contains at least three MBL sites. Higher latitudes (60-90°) are not considered in this comparison due to the gaps remaining in the OCO-2 $XCO_2$ record in high latitudes during wintertime and shouldering seasons.





**Table 2. Marine Boundary Layer stations within the NOAA Earth System Research Laboratory CO₂ sampling network**

| Region | Station | Acronym | Latitude | Longitude | Start Date | End Date |
|---|---|---|---|---|---|---|
| **Temperate Northern Hemisphere (20-60°N)** | Mace Head, Ireland | MHD | 53.3 | -9.9 | 2014-01 | 2020-07 |
| | Shemya, AK | SHM | 52.7 | 174.1 | 2014-01 | 2020-07 |
| | Terceira Island Azores | AZR | 38.8 | -27.4 | 2014-01 | 2020-07 |
| | Tudor Hill, Bermuda | BMW | 32.3 | -64.9 | 2014-01 | 2020-07 |
| | Sand Island, Midway | MID | 28.2 | -177.4 | 2014-01 | 2020-07 |
| | Key Biscayne, FL | KEY | 25.7 | -80.2 | 2014-01 | 2020-07 |
| **Tropical Northern Hemisphere (0-20°N)** | Cape Kumukahi, HI | KUM | 19.5 | -154.8 | 2014-01 | 2020-07 |
| | Mariana Islands, Guam | GMI | 13.5 | 144.7 | 2014-01 | 2019-08 |
| | Ragged Pointed, Barbados | RPB | 13.2 | -59.4 | 2014-01 | 2020-07 |
| | Christmas Island, Republic of Kiribati | CHR | 1.7 | 157.2 | 2014-01 | 2019-08 |
| **Tropical Southern Hemisphere (0-20°S)** | Seychelles | SEY | -4.7 | 55.2 | 2014-01 | 2020-07 |
| | Ascension Island | ASC | -8.0 | -14.4 | 2014-01 | 2020-07 |
| | Tutuila, America Samoa | SMO | -14.2 | -170.6 | 2014-01 | 2020-07 |
| **Temperate Southern Hemisphere (20-60°S)** | Cape Grim, Australia | CGO | -40.7 | 144.7 | 2014-01 | 2020-07 |
| | Baring Head | BHD | -41.4 | 174.9 | 2014-01 | 2020-07 |
| | Crozet Island | CRZ | -46.5 | 51.9 | 2014-01 | 2020-07 |


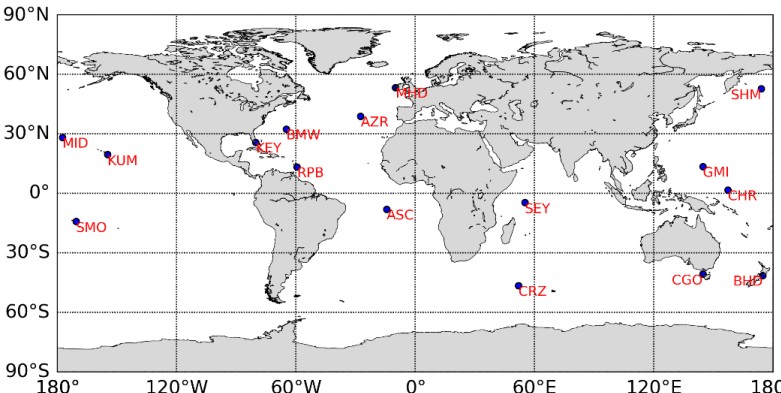

**Figure 2. Map showing the locations and the acronyms of the Marine Boundary Layer stations within the NOAA Earth System Research Laboratory CO2 sampling network.**

**2.1.4 Multivariate ENSO Index (MEI)**

We use the bi-monthly Multivariate El Niño/Southern Oscillation (ENSO) index (MEI; downloaded from Physical Sciences Laboratory: https://psl.noaa.gov/enso/mei/ ) to explore the relationship between CO₂ IAV and ENSO. The MEI is the time series of the leading combined Empirical Orthogonal Function of five different variables (sea level pressure, sea surface temperature, zonal and meridional components of the surface wind, and outgoing longwave

radiation) over the tropical Pacific basin. These ENSO-related variables are obtained from the Japanese 55-year



Reanalysis (JRA-55) (Kobayashi et al. 2015) and NOAA Climate Data Record of Monthly Outgoing Longwave Radiation. Positive values in the MEI indicate El Niño conditions, while negative values indicate La Niña conditions, and the magnitude reflects the relative strength. Unlike other ENSO indices which use only one climate metric (e.g., the sea level pressure difference between Tahiti and Darwin or the sea surface temperature anomaly within a pre-

defined box), the MEI provides for a more complete and flexible description of the ENSO phenomenon than traditional single variable ENSO indices and has less vulnerability to errors (Klaus Wolter et. al, 2011).

## 2.2 Methods

### 2.2.1 Spatial Aggregation

We aggregate daily $XCO_2$ observations from the version 10 OCO-2 Level 2 lite product to monthly scale, exploring patterns of IAV at three spatial scales: gridcell-level, zonal averages over 5° of latitude, and broad zonal belts. Aggregating soundings reduces random noise in the observations, mitigates the impact of data gaps due to cloud cover, and partly mitigates effect from low winter sunlight levels in polar regions. For gridcell level analysis, we aggregate data equatorward of 45° to 5°x5° bins since these data are not limited by polar night or degraded by high solar zenith

angles during winter. Poleward of 45° in both hemispheres, we aggregate the satellite observation to a latitude-longitude resolution of 5°x10° to compensate for fewer and noisier soundings in these latitudes, especially during winter and its shoulder seasons. Within each 5°x5° or 5°x10° gridcell, only months that have more than 5 soundings are included in the analysis. Our criteria for aggregation are based on sensitivity experiments in which we modulated the grid cell resolution from 1°x1° to 15°x15° (Fig. S1 and Fig. S2) and varied the threshold on the required number

of soundings within a month from 1 to 25 (Fig.S3, Fig.S4 and Fig.S5). Our goal was to reduce noise but maintain high spatial coverage (Fig.S6 and Fig.S7).

In our analysis, we also aggregate data to zonal averages. At intermediate spatial scales, we average all data around the 5° latitude bins described above. For comparison with TCCON and MBL data, which are spatially sparse, we

further aggregate $XCO_2$ data into four broad zonal belts – each of which contains at least 1 TCCON or 3 MBL stations -- (delineated in Table.1 and Table.2) to assess IAV patterns among the datasets. Keppel-Aleks et al., (2014) showed that drivers of IAV (i.e., temperature, drought stress, or fire) could be attributed when surface $CO_2$ were aggregated into similar broad zonal belts, whereas process-level attribution was not possible with global averaging. We therefore analyze broad zonal belts to gain a large-scale understanding of how three $CO_2$ datasets are similar and where

differences lie.

### 2.2.2 Deriving interannual variations

We use a consistent process to calculate IAV (Equation 1) from the raw OCO-2, TCCON and MBL timeseries. The

methodology is based on approaches used in Keppel-Aleks et al. (2013) and NOAA curve fitting methodology (Thoning et al, 1989 ). We decompose the raw time-series data into a long-term trend (which is a function of location





(x,y) and time (t)), a seasonal cycle (which is a function of location and calendar month (m)), and IAV anomalies using Equation 1:

$IAV(x,y,t) = Raw(x,y,t) − Trend(x,y,t) − Seasonal(x,y,m)$    Equation 1

We first fit a third order polynomial to the Raw timeseries to calculate the observed trend at each location (Fig. 3a). After removing the trend calculated at each gridcell (Fig. 3b), we calculate a mean seasonal cycle by taking the mean value of all January, February, etc. data (Fig. 3c). Particularly at high latitudes, some months are systematically under

sampled. For these gridcells, we must have at least two years with sufficient observations to calculate a climatological mean for that month, otherwise, that calendar month is assumed to have insufficient data to infer the IAV. Finally, we remove the mean seasonal cycle from the detrended timeseries at each gridcell to obtain the IAV anomaly timeseries (Fig. 3d). Given the short data record, we quantify the uncertainty in our calculation of the climatological seasonal cycle as the standard error for each calendar month (blue shading in Fig. 3c), and this uncertainty is propagated to the

corresponding IAV timeseries (Fig. 3d).

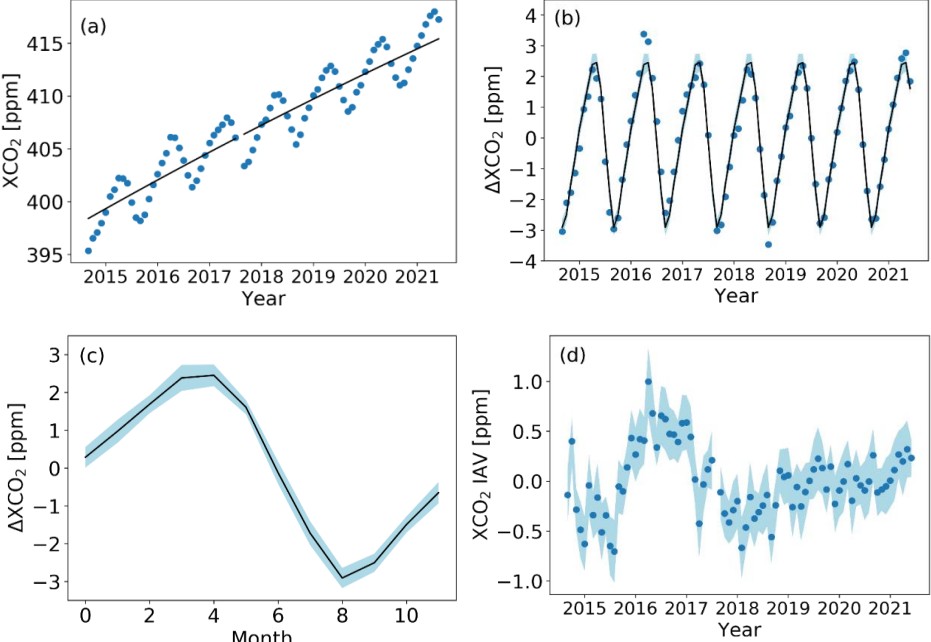

**Figure 3. Methodology to calculate the CO₂ interannual variability timeseries, using OCO-2 XCO₂ data at the 5-deg grid cell at 20°N, 155°W, which contains Moana Loa, as an example. (a) 5°resolution monthly mean raw OCO-2 XCO₂ and the associated 3rd order polynomial trend. (b) detrended monthly XCO₂ after removing the long-term trend with a repeating**

**12-month annual cycle obtained from calculating the mean for each month. The light blue shading gives the uncertainty of the seasonal cycle, which is derived by calculating the standard deviation across all Januarys, Februarys, etc. (c) 12-month mean annual cycle together with the uncertainty range plotted in (b). (d) Resulting interannual variability, when mean annual cycle is removed from detrended timeseries.**




## 3 Results

### 3.1 Temporal-Spatial Variations based on OCO-2 observations

When averaged into broad zonal belts representing the tropics and mid-latitudes, the OCO-2 XCO2 IAV timeseries anomalies range between -0.5 to 0.75 ppm (Fig. 4). All latitude bands show increasing IAV during positive MEI (El Niño) and decreasing IAV during negative MEI (La Niña), although the phasing varies among latitudes. During the strong 2015–2016 El Niño, which began around March 2015 and reached its peak at the start of 2016, XCO2 showed the largest IAV. The Southern Hemisphere extratropical regions have larger and more rapid response in the IAV associated with ENSO compared to other zones, especially for a second, smaller El Niño at the beginning of 2020 when the MEI peaked and the XCO2 IAV timeseries had an anomaly nearly twice as large as that of other latitude belts. During both El Niño events, the IAV timeseries in the NH tropics zone peaks nearly six months after the maximum MEI value.

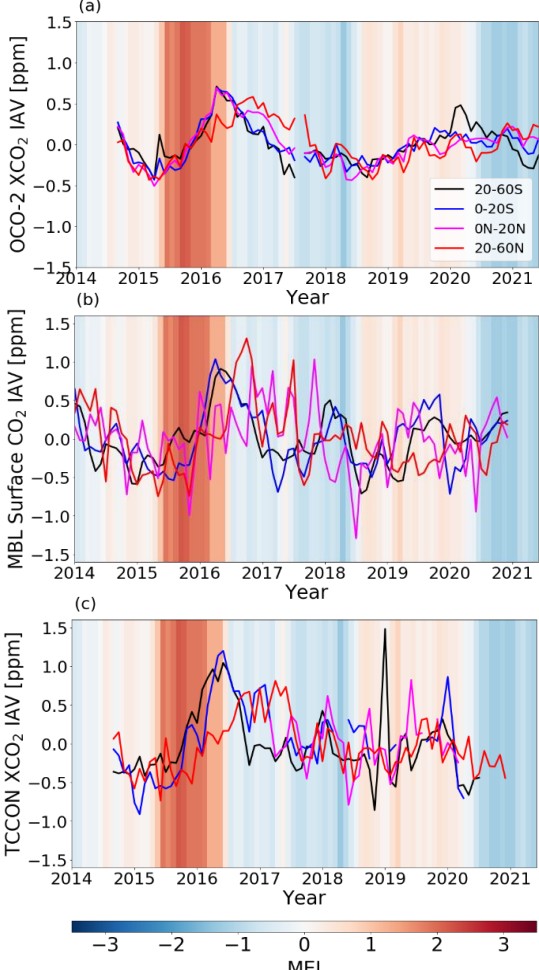

**Figure 4. IAV timeseries averaged for zonal bands between 60 °N and 60 °S from three different observing strategies. (a) Space-based OCO-2 XCO₂, (b) Surface CO2 observations from NOAA's marine boundary layer (MBL) sites, (c), Ground-**





The differences in temporal phasing between the broad zonal belts (Fig. 4a) associated with El Niño events can be

linked to transport of El Nino-driven $CO_2$ flux anomalies away from the tropics when zonal means are calculated from

OCO-2 observations at 5° latitude resolution (Fig. 5). For the two El Nino periods in 2015-2017 and late-2018 to

2021, high IAV values originate in the tropics and a smooth transition to high IAV values is seen at higher latitudes

as time progresses (Fig. 5a). We note that fluxes outside the tropics may also be influenced by ENSO-related climate

variability, yet the transport of tropical-driven anomalies appears to dominate. This 7-year study period also captures

the half-year lags for atmospheric transport or climate-ecological teleconnections that impacts $XCO_2$ variations in the

far North.

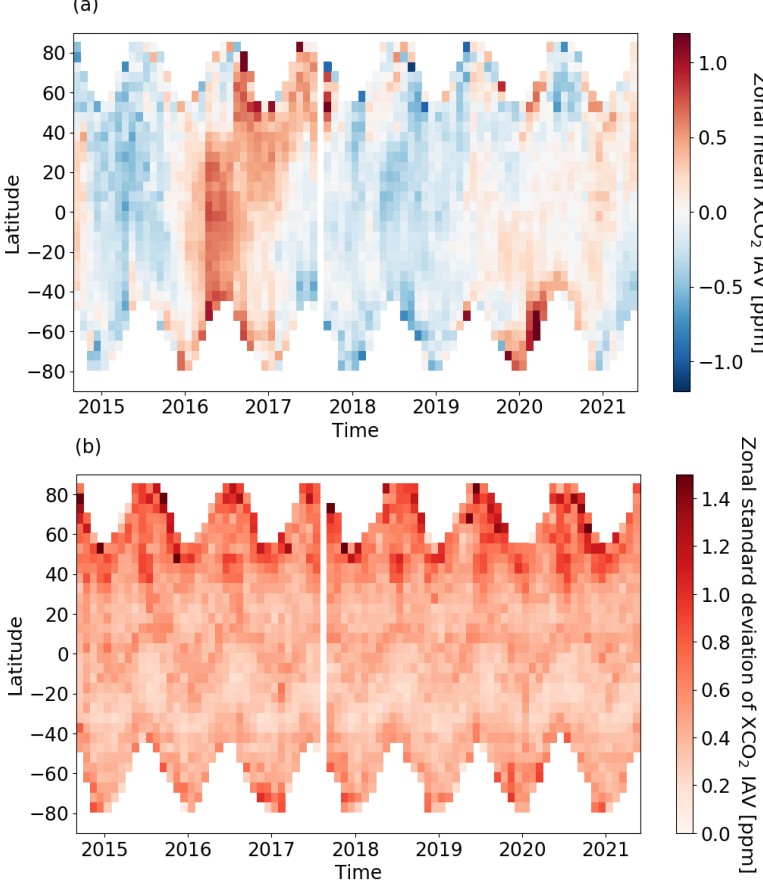

**Figure 5. Hovmöller Diagrams diagram showing zonal mean OCO-2 XCO₂ IAV timeseries for 5° latitude bins (a) and the**
**zonal standard deviation of XCO₂ IAV (b), which gives an estimate of coherence in the IAV patterns among grid cells in**
**the 5° zonal belt.**



We quantify coherence in $CO_2$ IAV within a latitude circle by taking the standard deviation across gridcell-level IAV anomalies within each 5° latitude zone. The standard deviation among gridcells is highest in the far North, with values as high as 1 ppm poleward of 45°N and as low as 0.2 ppm in the Southern Tropical bands (Fig. 5b), indicating that IAV is less spatially coherent in the Northern Hemisphere. This may be consistent with studies that show greater IAV in terrestrial ecosystem fluxes (concentrated in the northern hemisphere) (Zeng et al., 2005) relative to ocean fluxes, or may reflect that our IAV timeseries also retains the imprint of sampling, measurement, and retrieval errors, which become more pronounced at higher latitudes. In general, there is not a time-dependent or ENSO-related pattern for the longitudinal variation of IAVs (no obvious changes during the two El Nino periods), which suggests the variation within each 5° band may be approximately stable and does not change substantially with interannual climate events.

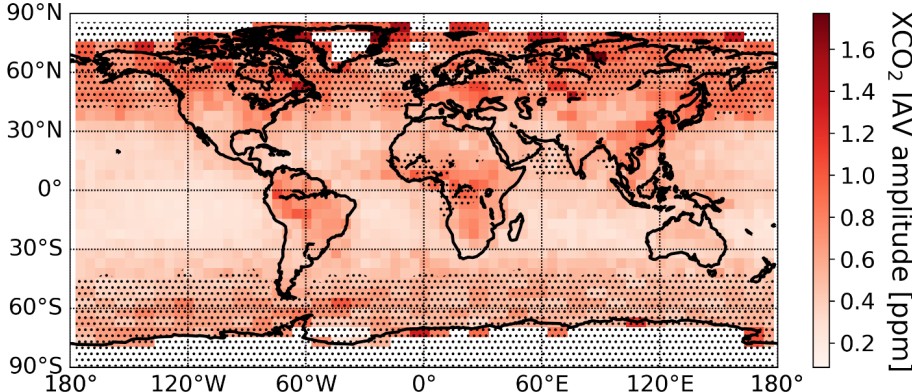

**Figure 6. XCO₂ IAV amplitude, determined as the standard deviation of the IAV timeseries. Data equatorward of 45° are averaged at 5°by 5° resolution , and data poleward of 45° are averaged at 5°by 10° resolution. Shaded regions indicate gridcells that lack mean annual cycle data for at least two calendar months due to polar night or related retrieval challenges.**

The XCO₂ IAV amplitude (the standard deviation of the IAV timeseries) is notably larger over continental gridcells compared to ocean gridcells (Fig. 6). In both hemispheres, the IAV amplitude over subtropical ocean basins is less than 0.4 ppm, while the IAV amplitude over tropical land in Southeast Asia, Congo forests and Amazon Basin is about 1 ppm. In higher latitudes, the XCO₂ IAV amplitude can exceed 1.2 ppm above deciduous and boreal forests in North America and Eurasia. Higher values over land likely occur due to the active $CO_2$ exchange between terrestrial ecosystem and the atmosphere, but we cannot rule out that retrievals over land show more variance due to complex topography, albedo, etc., which are elements of the retrieval state vector. Nevertheless, over land areas with low carbon exchange (e.g., Australia, the Middle East, the Sahara Desert), the XCO₂ IAV amplitude is nearly of the same low level as the ocean basins. It is worth noting that for high latitude regions, including both Northern continents and Southern Ocean, OCO-2 does not obtain observations over a full calendar year (stippled gridcells in Fig. 6) due to polar nights, low light levels, and high solar zenith angles. The XCO₂ IAV amplitudes are less zonally coherent

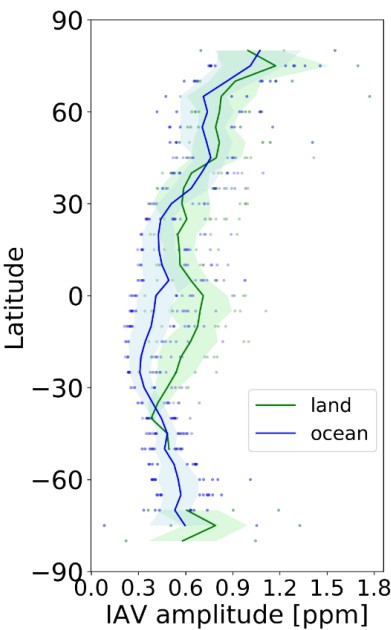

**Figure 7. Latitudinal profile for zonal mean of IAV amplitude and the standard deviation among land (green) or ocean (blue) gridcells in each latitude band (shaded area). Individual points represent all grid cells valid IAV record within the certain zonal band.**

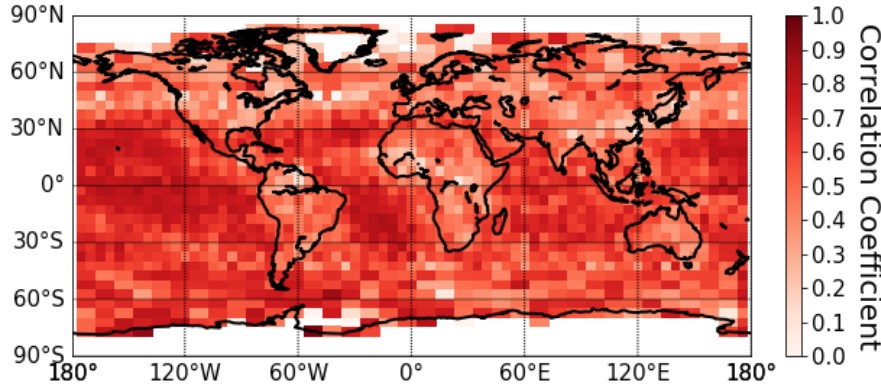

**Figure 8. Correlation coefficient between local grid cell IAV timeseries and the corresponding 5° zonal mean XCO₂ IAV timeseries.**

through these regions than those in the tropics and mid-latitudes, for both land and ocean. When averaging all ocean or land grid cells around a latitude circle, the zonal mean IAV amplitude over the ocean ranges from 0.3 to 1.0 ppm, while the land IAV amplitude ranges from 0.4 to 1.1 ppm (Fig. 7). Both the land and ocean profiles have similar north-south patterns, with higher IAV amplitude in the Northern Hemisphere and lower IAV amplitude in the Southern Hemisphere, and the small IAV amplitudes in the subtropics of both hemispheres, with more scatter among land





gridcells than ocean (Fig. 5b and  Fig. 7), suggesting either the influence of local flux IAV on land or greater error
        associated with retrievals on land. We note better coherence between the $XCO_2$ IAV timeseries of each local grid cell
        and that of zonal mean $XCO_2$ IAV timeseries for ocean, with correlation coefficients of approximately 0.8. In contrast,
        land gridcells are generally correlated with the zonal mean at around 0.4 to 0.6 (Fig. 8).

**3.2  $XCO_2$ IAV compared to surface and TCCON ground-based sites**
        Given that the small IAV signal (up to 1 ppm over land, and smaller over ocean) is similar in magnitude to noise and
        systematic bias in OCO-2 soundings (Torres et al., 2019), we corroborate patterns of IAV from OCO-2 with other
        datasets. The OCO-2 IAV timeseries in broad latitudinal belts share similarities with those of TCCON $XCO_2$ and
        MBL surface $CO_2$ IAV timeseries, with all timeseries showing similar relationships to MEI. Especially striking is that
all timeseries capture the lagged response in the NH midlatitude belt to the strong 2015/16 El Nino (Fig. 4a-c).
        Although the patterns are similar, the magnitude of IAV at the MBL sites is almost double the IAV in the OCO-2
        $XCO_2$ timeseries. Given that the atmospheric boundary layer, where surface observations are made, is on average 10%
        of the total column, this suggests that much IAV in total column observations is present within the free troposphere.
        For TCCON, the amplitude of IAV is similar to that of OCO-2, since both methods capture total column variations.
We note that the zonal IAV timeseries for MBL and TCCON appear to have more high frequency variations than those
        from OCO-2 (Fig. S11 & S12), which likely stems from the fact that the zonal composites are developed from sparse
        ground-based sites (between 1 and 12 observatories) within each latitude belt, whereas the satellite measures at all
        longitudes within a belt, though with more limited time resolution. The zonal mean OCO-2 observations are correlated
        with MBL sites within the same latitude band with R between 0.5 and 0.75 (diagonal elements on Fig. 11b).
Correlations between zonal TCCON and OCO-2 observations range between 0.15 and 0.55 (Table. S1). The
        correlations are weakest in the northern tropics band, where TCCON data were unavailable during the strong El Nino
        (Fig. 3c). It is noteworthy that OCO-2 zonal averages are more correlated among different latitudes than are MBL or
        TCCON observations (off-diagonal elements in Fig. 11c, d, e). The greater correlation across latitudes for OCO-2
        compared to MBL sites is likely due to the sensitivity of the OCO-2 $XCO_2$ observations to the free troposphere, where
meridional transport is more rapid than at the surface. While TCCON data are also sensitive to the free troposphere,
        we hypothesize that the zonal belt averages for TCCON, constructed from only a few sites, are more affected by noise,
        both instrumental and geophysical, and thus show lower coherence than the OCO-2 $XCO_2$ averages constructed from
        the whole latitudinal bands.

We further compared the IAV from OCO-2 $XCO_2$ with TCCON stations at the site level (Fig. 10). Across all sites,
        the IAV amplitude generally shows good agreement and lies between 0.4 to 1.2 ppm. We note a slight low bias in
        OCO-2 relative to TCCON for all five sites in the Southern Hemisphere which lie below the one-to-one line. Low
        OCO-2 IAV amplitudes may be due to the fact that a 5x5 ° gridcell encompassing these near-coastal locations includes
        both land and ocean OCO-2 soundings, and may be due to specific sources of variance from retrieval bias affected by
surface type for the OCO-2 (e.g., Fig. 7). It is also worth noting that OCO-2 is looking at a region of 5° by 5° gridcell



(or 5° by 10° in higher latitudes) around TCCON sites, so there are different signals affecting the variance between the two types of observations.

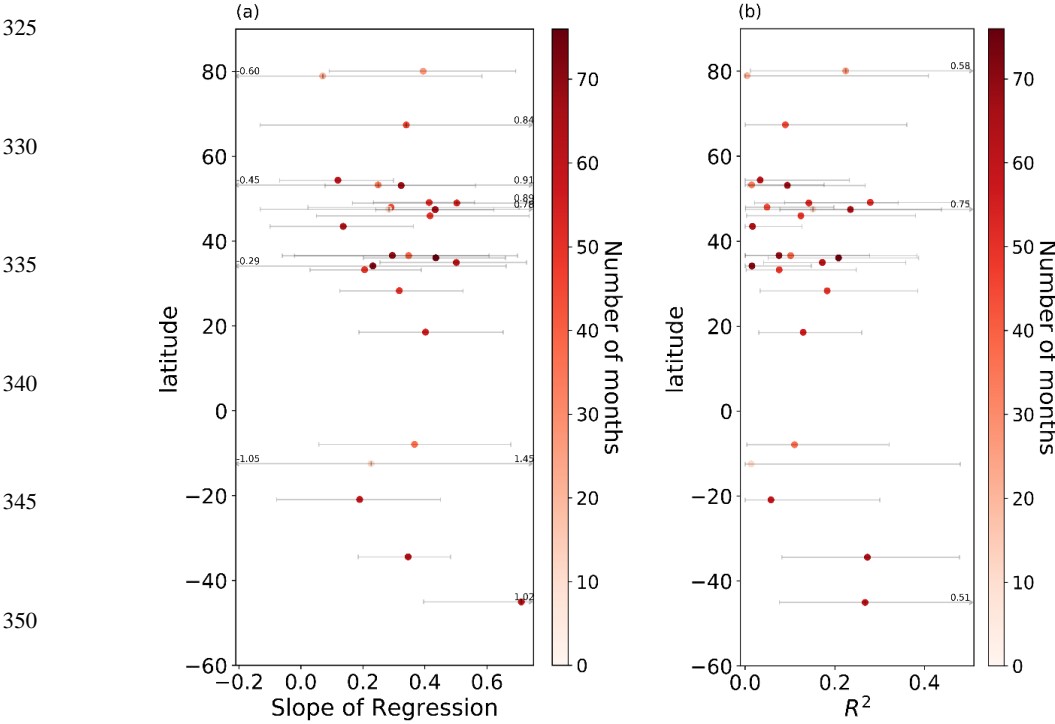

**Figure 9. Latitudinal profile of regression Slope (panel a) and correlation coefficient (R, panel b) of OCO-2 versus TCCON XCO2 IAV. The Slope and R values are based on using monthly XCO₂ IAV. The error bars result from a Monte Carlo bootstrapping approach . The colours represent the number of months data which used for the regression calculation, given gaps in both the OCO-2 and TCCON datasets.**

We derive the regression slopes and R between OCO-2 and monthly averaged TCCON IAV through bootstrapping Linear Regression fitting techniques to investigate the coherence between IAV signals from space-based and in-situ ground-based observations. We compute the linear regression 1000 times, by iteratively resampling the IAV timeseries with replacement, and calculate the 95% significant level for regression slopes based on the histogram of the sample distributions during the bootstrapping (Fig. S10). Despite having similar IAV amplitudes, the IAV timeseries from OCO-2 are only moderately correlated with those from TCCON (Fig. 9). The regression slopes range from 0.1- 0.6 and R values are generally around $0.1 - 0.5$, indicating that less than 25% of the IAV in OCO-2 is explained by IAV measured by TCCON. These R values are, as expected, smaller than the zonal averages shown in Fig. 11b, which average some of the site-level noise for TCCON and gridcell-level noise for OCO-2. The detailed XCO₂ IAV



timeseries of each site (Fig. S11) for OCO-2 and TCCON show that the IAV timeseries in the NH are more variable, which can partly explain the hemispheric difference in amplitude, slope, and correlation coefficients.

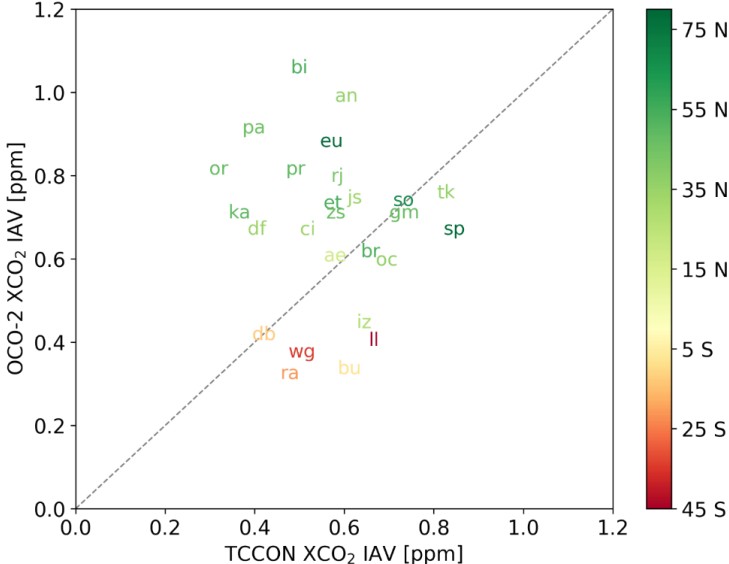

**Figure 10. Comparison of OCO-2 and TCCON XCO₂ IAV amplitude at individual sites. Colours reflect site latitudes. The grey dashed line is the one-to-one identity line.**

## 4 Discussion

We use seven years of OCO-2 total column carbon dioxide observations from late 2014 to mid-2021 to illustrate the global temporal-spatial patterns of atmospheric $XCO_2$ interannual variations. The good agreement among the OCO-2 $XCO_2$ IAV timeseries, TCCON $XCO_2$ IAV timeseries and the MBL surface $CO_2$ IAV timeseries in broad zonal belts improves our confidence that we are able to quantify reasonable IAV timeseries from the satellite record. From the space-based and ground-based detection, we are able to characterize the global response of OCO-2 and TCCON $XCO_2$ or MBL surface $CO_2$ IAV to ENSO, and track the $CO_2$ IAV against the positive/negative phase of ENSO, together with the transport of the signal from South to North (Fig. 4). All datasets show consistent patterns in the response to the El Nino periods, although we note that the IAV amplitude is a factor of almost two smaller in the column average mole fraction, which reflects the fact that IAV variations emerge due to surface fluxes in the lower part of the atmosphere (Olsen & Randerson, 2004), but are efficiently transported into the free troposphere which comprises the bulk of the column. When taken together, the use of surface and column data may allow better separation of transport-driven versus local flux driven variations at the interannual timescale.



**Figure 11.** Correlation coefficient (R) between among mean CO₂ timeseries using three observing strategies. Panel (a) shows the correlation between zonal mean OCO-2 XCO₂ IAV and zonal mean marine boundary layer CO₂. Panel (b) shows the correlation between zonal mean XCO₂ IAV from OCO-2 and TCCON. Panels (c-e) show the correlation in zonal mean IAV timeseries across four latitude bands for a single observing strategy. Panel (c) shows OCO-2 XCO₂, Panel (d) shows MBL CO₂ and Panel (e) shows TCCON XCO₂. For Panels c-e, the diagonal elements are 1 by construction. Zonal bands include tropical (0-20 degree) and NH/SH temperate zone (20-60 degree)



Our results, however, underscore the difficulty in detecting IAV signals from remote sensing of $XCO_2$ -- while northern hemisphere seasonal amplitudes are typically 10 ppm scale (Basu et al., 2011), the magnitude of OCO-2 detected $XCO_2$ IAV is almost an order of magnitude smaller (less than 0.4 ppm over ocean and about 1ppm over continents). The magnitude of IAV is therefore comparable to other components of the $XCO_2$ variance budget; for

instance, Torres et al. (2019) show random noise in individual OCO-2 soundings of about 0.3 ppm in the southern hemisphere and of about 0.7 ppm in the northern hemisphere, and spatially coherent errors in the retrievals ranging from 0.3 to 0.8 ppm (Torres et al., 2019). Moreover, the uncertainty which originally comes from the varying climatological seasonal cycle, can also reach the level of 0.5ppm (Fig. 3d). Therefore, robust partitioning of IAV from the observed $XCO_2$ signal at a given location requires a comprehensive variance budget (Mitchell et al., submitted),

and efforts to infer interannual variations in fluxes from OCO-2 must take gridcell-level variance into account or leverage zonally averaged data, which is characterized by greater separation between IAV signal and noise.

Our analysis shows that proper spatial averaging of the monthly $XCO_2$ signal can mitigate the imprint of random noise and systematic effects from weather systems at sub-monthly timescales. Based on sensitivity tests, we recommend

averaging low to mid-latitude of $XCO_2$ (equatorward of 45°) to 5°x5° bins, and 5° latitude x 10° longitude grid cell poleward of 45°, ensuring that each gridcell aggregates at least 5 soundings within a month. At these levels of spatial averaging, the $XCO_2$ IAV amplitude was comparable to that of the co-located ground based $XCO_2$ IAV amplitude measured by TCCON (Fig.10). However, the moderate to low correlation between the IAV timeseries from each monitoring platform reveals the discrepancies of the two measurements in sampling, detection or retrieval, suggesting

that one or both is still convolving another source of variance with the calculated IAV signal. Based on the good agreement between the two timeseries in broad zonal belts, we expect that random noise in both observations may degrade the comparison.

The smaller coherence in the IAV timeseries in nearby land and ocean gridcells may be due to larger error over land

or may reflect that $XCO_2$ observations over land contain information about heterogeneous local flux IAV. Complete analysis of the variance budget for OCO-2 observations (Mitchell et al., submitted) will elucidate the likely imprint of each process. When using IAV timeseries for flux inference, it will be crucial to account for non-flux imprints on the timeseries, since spurious attribution of IAV will lead to biased fluxes.

**5 Conclusions**

We examined IAV in OCO-2 data to determine whether the small variations that result from interannual flux variations can be detected in light of other sources of variance in the space-based dataset. Our results show that zonal averages reveal relationships with ENSO that are consistent with those from established ground-based monitoring network. Zonal averages greatly reduce random noise in $XCO_2$ compared to 5x5° averages. In general, OCO-2 can successfully

monitor $CO_2$ IAV over both land and ocean, contributing important spatial coverage beyond inferences of IAV from existing ground-based networks.



**Data availability**

The version 10 OCO-2 Level 2 bias-corrected XCO2 data product is available from Goddard Earth Sciences Data and

Information Services Center Archive: https://disc.gsfc.nasa.gov/datasets/OCO2_L2_Lite_FP_10r/summary. TCCON

Data is publicly available from the TCCON Data Archive (https://tccondata.org/) hosted by the California Institute of

Technology. MBL dry air mole fraction data is available from NOAA Global Monitoring Laboratory Earth System

Research Laboratories Archive: https://gml.noaa.gov/ccgg/mbl/data.php.

**Author contribution**

Formal analysis: Yifan Guan. Writing – original draft preparation: Yifan Guan. Conceptualization: Gretchen Keppel-

Aleks. Supervision: Gretchen Keppel-Aleks. Project administration: Gretchen Keppel-Aleks and Scott C. Doney.

Writing – review & editing: Gretchen Keppel-Aleks, Scott C. Doney, Christof Petri, Dave Pollard, Debra Wunch,

Frank Hase, Hirofumi Ohyama, Isamu Morino, Justus Notholt, Kei Shiomi, Kim Strong, Kivi Rigel, Matthias

Buschmann, Nicholas Deutscher, Paul Wennberg, Ralf Sussmann, Voltaire A. Velazco and Yao Té.

**Competing interests**

The authors declare that they have no conflict of interest.

**Acknowledgements**

The authors thank the participants of the NASA OCO-2 mission for providing the OCO-2 data product (From GES

DISC Archive: https://disc.gsfc.nasa.gov/datasets/OCO2_L2_Lite_FP_10r/summary) used in this study. We thank

TCCON partners for providing total column data. We acknowledge NASA support through the OCO Science team

and grants 80NSSC18K0897, 80NSSC21K1071, 80NSSC18K0900, and 80NSSC21K1070 to the University of

Michigan and University of Virginia. The Paris TCCON site has received funding from Sorbonne Université, the

French research center CNRS, the French space agency CNES, and Région Île-de-France. TCCON sites at Tsukuba,

Rikubetsu and Burgos are supported in part by the GOSAT series project. Burgos is supported in part by the Energy

Development Corp. Philippines, the TCCON site at Reunion Island has been operated by the Royal Belgian Institute

for Space Aeronomy with financial support since 2014 by the EU project ICOS-Inwire and the ministerial decree for

ICOS (FR/35/IC1 to FR/35/C6) and local activities supported by LACy/UMR8105 and by OSU-R/UMS3365 –

Université de La Réunion. The TCCON stations at Garmisch and Zugspitze have been supported by the Helmholtz

Society via the research program "Changing Earth – Sustaining our Future". The Eureka TCCON measurements were

made at the Polar Environment Atmospheric Research Laboratory (PEARL) by the Canadian Network for the

Detection of Atmospheric Change (CANDAC), primarily supported by the Natural Sciences and Engineering

Research Council of Canada, Environment and Climate Change Canada, and the Canadian Space Agency.



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
