# Peer review of "Characteristics of Interannual Variability in Space-based XCO2 Global Observations"

_EGUsphere, 2022_

## Referee Comment (RC1)

Review of
"Characteristics of Interannual Variability in Space-based XCO$_2$ Global Observations"
By Guan et al.
Submitted to ACP

*Summary*

This manuscript evaluates the interannual variability (IAV) of OCO-2 XCO2 observations over the period 9/2014-12/2020.  Specifically, the IAV of the detrended, deseasonalized XCO2, which can be thought of roughly as "IAV of the XCO2 growth rate" (which is how this paper describes it).  The manuscript shows that broad features of the IAV on broad zonal scales are strongly correlated with ENSO phases, and further how the XCO2 increases caused by the 2015-2016 El Nino propagate from the tropics to the northern extratropics over a period of roughly 6 months.  They compare IAV from OCO-2 with that from 26 TCCON stations (individually and aggregated by latitude band) as well as boundary layer CO2 from select NOAA surface stations.  They find general agreement from all three data sources, though data at individual locations are very noisy, regardless of data source.

Overall, I find the paper very well-written and generally complete.  I think it will be an interesting addition to the literature, further advocating use of space-based XCO2 data in different ways.  However, I do find some small deficiencies throughout the paper. Therefore, I recommend publication after dealing with the various (mostly minor) issues & questions I raise below.

*General Comments*

So much of the abstract seems, well, abstract.  Is this a validation paper, basically validating OCO2 IAV so we can have more confidence in flux inversion results?  Or to better understand the spatiotemporal scales at which we should aim our flux efforts which utilize OCO2 data?  I suggest making the abstract a bit more clear about how much the manuscript is validating OCO2 IAV, versus to what degree it is doing interesting analysis with the IAV itself.

Regarding comparisons to TCCON & MBL sites: it seems like because of the low sampling associated with TCCON and the MBL sites, OCO-2 derived IAV is more powerful because of the better spatial sampling.  You may wish to point this out in the abstract and/or conclusions more specifically.   (I also wonder how much better future wide-swath sensors may be).  Did you ever consider applying your method to GOSAT to derive IAV, to see how it compares to OCO-2?  It would be especially interesting as we have 11+ years of GOSAT XCO2.

*Specific Comments*

Abstract: *"The amplitude of IAV variations is up to 1.2 ppm over the continents and around 0.4 ppm over the open ocean."* Please make it clear in the abstract that you are defining "amplitude of" as "standard deviation of".

Sec 2.2.1: The results of the spatial aggregation sensitivity analysis seem to show substantial differences in the IAV depending on the spatial scale of aggregation. How do you know which spatial scale is most accurate, given that the differences are not just noise, but show large-scale biases? These large-scale differences clearly matter, as you show later most of the IAV is less than 0.75 ppm. I strongly suggest you repeat your sensitivity analysis with high-resolution model data (rather than real data), sampled like OCO-2. If you use model data, you know the right answer, so you can see what you can get away with. Something like the GMAO 0.75 deg model should have sufficient resolution for this purpose.

Sec 2.2.2: It seems like using a 3$^{rd}$ order polynomial on a 7-year time series to remove the secular increase is a recipe for problems when trying to derive the IAV. Wouldn't this artificially remove some of the IAV? Please discuss why 3$^{rd}$ order is necessary in the paper. Did you test 1$^{st}$ or 2$^{nd}$ order ? If so, why were they not sufficient?

Fig 5a: Care to comment on the strong feature near the beginning of 2020 peaking at 60S latitude? That seems stronger that random variability.

Near Fig6: Because MEI/ENSO is such a heavily discussed topic in this work, a plot of the correlation coefficient of MEI with IAV timeseries in local 5x5 gridboxes may be warranted – similar to figure 6. Have you made such a plot, and does it show any interesting teleconnections? You may need to introduce a lag at the more northern latitudes when calculating correlation coefficients there (a simple 0-6 month lag as a function of latitude could work).

Figure 10: Each "point" on the plot has in fact some uncertainty on the IAV at each site, due to both retrieval errors and spatiotemporal noise. Is it possible to get an estimate of this, and use it to add x & y error bars on each point? That might give a better picture of how consistent TCCON and OCO-2 IAV are, to within their respective errors. This figure implies that they are not very consistent.

Related to the above, please check your IAV stddev calculations. I tried to reproduce your Bialystok numbers for OCO2 and TCCON. Just by eyeballing your Fig S11, I got 1.05 for OCO2 (similar to your number), but I got 0.75 for TCCON, whereas you got roughly 0.5. I wonder if some of your TCCON values are too low for some reason (in particular at the NH sites). Your values at Karlsruhe and Orleans also both seem unreasonably low (both less than 0.4 ppm).

*Technical Comments*

Line 72: Remove comma after "Chatterjee et al."

L76: "…is being used implicitly for flux attribution…". Please provide example references.

L85: Please add reference Baker et al., *Geosci Mod. Dev.,* https://doi.org/10.5194/gmd-15-649-2022.

L110: Replace the O'Dell et al, 2012 reference with the O'Dell et al, 2018 reference. The former applies to GOSAT; the latter applies to OCO-2 and is a much more appropriate reference.

Sec 2.1.2: Please state somewhere if you use GGG2014 or GGG2020 (I'm assuming the former).
Sec 2.1.3: If there is any kind of version number or data source website for the NOAA sampling data, please provide it.

Figure 1: Most of the TCCON sites are in completely wrong places!! It looks like the longitudes are screwed up?

L161: Please provide a reference for the NOAA monthly OLR data set, or remove the sentence about the source of the ENSO-related variables (which is probably not necessary as it will be given in the MEI documentation).

Figure 9 caption: Do you show R or $R^2$ in panel b? Please make the caption more clear. Currently the caption says R and the plot says $R^2$ on the axis label.

Line 362: Change "R" to "correlation coefficients R". Otherwise we really have to guess that you mean correlation coefficient.

Line 387: "although we note that the IAV amplitude is a factor of almost two smaller in the column average mole fraction". Relative to what? Please add "relative to boundary layer $CO_2$" or something similar.

Fig S1 Caption: Please give the min # of soundings per gridbox.
Fig S3 Caption: Please give the spatial gridding (5x5, etc) used, and change word "use" to "using" in the caption.

Fig S6 Caption: Suggest changing word "record" to "years" ? It took me a while to figure out what you were getting at here.

---

## Referee Comment (RC2)

**Interactive discussion on "Temporal-Spatial Variations based on OCO-2 observations" by Yifan Guan**

**Summary:** This manuscript assesses the IAV (interannual variability) of the column-averaged carbon dioxide ($CO_2$) dry air mole fraction of OCO-2 satellite data (ACOS version 10) at a global scale for September 2014 to June 2021 and associates its variability to El Niño/Southern Oscillation (ENSO) conditions. The IAV assessment was performed at different latitudinal bands (20-60S, 0-20S, 0N-20N and 20-60N) and a grid-cell spatial scale of 5x5 degrees. The main finding suggests that positive OCO-2 anomalies in 2015-2016 are associated with ENSO conditions, and similar results are found when analysing TCCON and MLB observations. Although OCO-2 IVA patterns are similar to based-ground observations, the amplitude of the IVA is much larger for MBL sites and TCCON.

In general, the manuscript is organized and well-written. However, some parts in the results sections need clarification. Therefore, I would recommend this paper for publication once the authors have addressed all the questions described below.

General comments:

Given that the IVA at MBL sites is almost double the IAV in the OCO-2 time series (Fig. 4a and 4b), Has the author tried to estimate XCO2 at the lower layer of the troposphere? Some studies (e.g., Kulawik et al., (2017)) split the XCO2 GOSAT column into two partial columns (lower <25 km elevation) and above 25 km., and found that the lower tropospheric CO2 partial column compares well to the independent surface CO2 data at oceanic sites. I wonder whether doing something similar here could improve the IVA comparison between OCO-2 and MLB. I also agree with reviewer 1, and a comparison with GOSAT XCO2 satellite data will also be beneficial.

Specific comments:

Line 221: When averaged into broad zonal belts representing the tropics and mid-latitudes, the OCO-2 XCO2 IAV time-series anomalies range between -0.5 to 0.75 ppm (Fig. 4). Does the author mean Fig.4a? Please indicates the results of Figs 4b and 4c are described in 3.2. For a moment, I thought that author was going to include the findings of MLB and TCCON in this section.

As an opinion, I believe that it would be better to only have one figure in this section (Fig 4a), and later in Section 3.2, I would include Fig 4a-4c as 4 different panels. For example, (a) (20-60S), (b) 0-20S, (c) 0N-20N and (d) 20-60N, where each panel should only contain the IVA time-series of OCO-2, MLB and TCCON together. By doing this, it would be easy for the reader to see how different the temporal-spatial variation is between these products.

Line 225: The Southern Hemisphere extratropical regions have larger and more rapid response in the IAV associated with ENSO compared to other zones, especially for a second. What does the author mean by saying: especially for a second?

The author also mentions that TCCON has a similar IAV amplitude to OCO-2. Is this true? Looking at Fig 4. c, TCCON has more IVA than OCO-2 (zonal belt (20-60S)). What happened in 2019? It seems that TCCON has an IAV of about 1.5 ppm compared to OCO-2 (no variability).

We note a slight low bias in OCO-2 relative to TCCON for all five sites in the Southern Hemisphere, which lie below the one-to-one line. How does the author know that OCO-2 has a lower bias than TCCON? I am a bit confused here; the author calculated standard deviations in Fig.10 and then discussed biases. Please clarify.

Line 427: When using IAV time series for flux inference, it will be crucial to account for non-flux imprints on the time series since spurious attribution of IAV will lead to biased fluxes. What does the author mean when he says: non-flux imprints on the time series? Flux or XCO2? Spurious attribution of IAV in XCO2 data?

Editorial comments:

Table 1 and Table 2 should also be included in the appendix and not in the main text.

Please be aware that TCCON locations must be placed correctly on the map. For example, reunion Island is over the Australian continent.

Line 227: that of other latitude belts. Remove 'that of' from the text; it seems unnecessary in this sentence.

EL niño instead of El nino. Please be aware that el niño is a Spanish word that must be written with 'Ñ'.

**References**

Kulawik, S. S., O'Dell, C., Payne, V. H., Kuai, L., Worden, H. M., Biraud, S. C., Sweeney, C., Stephens, B., Iraci, L. T., Yates, E. L., and Tanaka, T.: Lower-tropospheric CO2 from near-infrared ACOS-GOSAT observations, Atmos. Chem. Phys., 17, 5407–5438, https://doi.org/10.5194/acp-17-5407-2017, 2017.

---

## Author Comment (AC1)

**General Comments**

*Reviewer's comments:*

*"So much of the abstract seems, well, abstract. Is this a validation paper, basically validating OCO-2 IAV so we can have more confidence in flux inversion results? Or to better understand the spatiotemporal scales at which we should aim our flux efforts which utilize OCO-2 data? I suggest making the abstract a bit more clear about how much the manuscript is validating OCO-2 IAV, versus to what degree it is doing interesting analysis with the IAV itself.*

*Regarding comparisons to TCCON & MBL sites: it seems like because of the low sampling associated with TCCON and the MBL sites, OCO-2 derived IAV is more powerful because of the better spatial sampling. You may wish to point this out in the abstract and/or conclusions more specifically. (I also wonder how much better future wide-swath sensors may be). Did you ever consider applying your method to GOSAT to derive IAV, to see how it compares to OCO-2? It would be especially interesting as we have 11+ years of GOSAT $XCO_2$."*

Authors' Response:

We thank the reviewer for their generally positive review and for the constructive suggestions. We agree with the reviewer, there are many elements to the analysis we are presenting, and the paper consists of two parts: first, validating the usage of space-based OCO-2 detection by comparing the $XCO_2$ IAV based on the OCO-2 satellite against TCCON ground-based and MBL sites observation. Second, characterizing the spatiotemporal patterns of the interannual variation of the atmospheric $CO_2$ to understand its global drivers, since OCO-2 provides better spatial sampling than past datasets. In response to the reviewer's suggestion, we incorporate GOSAT observations available since 2009 in our analysis. GOSAT shows the similar pattern as OCO-2, and we are able to see that OCO-2 data show improvements in terms of the data quality, as expected.

Following the reviewers' suggestion, we add a bit more explanation for the purpose of the paper in the abstract/conclusion sections, and we describe the benefits of the improved spatial coverage of OCO-2. We also added figures and analysis based on the GOSAT dataset.

Key new lines in the abstract read "The similar zonal patterns of OCO-2 $XCO_2$ IAV timeseries compared to ground-based in situ observations and with column observations from the Total Carbon Column Observing Network (TCCON) and the Greenhouse Gases Observing Satellite (GOSAT) provide validation that OCO-2 observations can be used reliably to estimate IAV. Furthermore, the extensive spatial coverage of the OCO-2 satellite data leads to more robust IAV timeseries than those from other datasets, suggesting that OCO-2 provides new capabilities for revealing small IAV signals despite sources of noise and error that are inherent to remote sensing datasets. "

**Specific Comments**

*Reviewer's comments:*

*Abstract: "The amplitude of IAV variations is up to 1.2 ppm over the continents and around 0.4 ppm over the open ocean." Please make it clear in the abstract that you are defining "amplitude of" as "standard deviation of".*

 Authors' Response:

We added explanation that amplitude is calculated as the standard deviation of the timeseries. The revised abstract reads "The IAV amplitude, calculated as the standard deviation of the IAV timeseries, is up to 1.2 ppm over the continents and around 0.4 ppm over the open ocean."

*Reviewer's comments:*

*Sec 2.2.1: The results of the spatial aggregation sensitivity analysis seem to show substantial differences in the IAV depending on the spatial scale of aggregation. How do you know which spatial scale is most accurate, given that the differences are not just noise, but show large-scale biases? These large-scale differences clearly matter, as you show later most of the IAV is less than 0.75 ppm. I strongly suggest you repeat your sensitivity analysis with high-resolution model data (rather than real data), sampled like OCO-2. If you use model data, you know the right answer, so you can see what you can get away with. Something like the GMAO 0.75 ° model should have sufficient resolution for this purpose.*

Authors' Response:

We agree with the reviewer that conducting a sensitivity analysis with high-resolution model data such as GMAO 0.75° would be interesting, yet we consider that the work involved in the suggested model analysis would first require a validation of the model IAV, and the scope of adding a model analysis is beyond the scale of our current study. Therefore, we prefer to keep this paper focused on observations only.

To address the reviewer's concerns, we conducted sensitivity studies with the observations themselves in which we aggregate the OCO-2 detection in different resolution - from 1° resolution to 15° resolution, and compare the difference of these resolutions (In Fig S1 and S2) to find the threshold that balanced the two goals of reducing noise yet revealing IAV at sub-zonal resolution. At 5°x5° (lower latitudes and 5°x10° (higher latitudes), the appearance of hotspots is minimized and the IAV amplitudes are spatially smooth, which we interpret as IAV signals emerging above the noise.

We further calculated the correlation coefficient among the IAV timeseries in neighboring gridcells of 1° resolution (Fig. S9). The R value decreases as the distance between the gridcell increases in each 10° zonal bands from 70°N to 70°S. We see a rapid decrease in the correlation between gridcells separated by 1° and gridcells separated by 5°, and generally stable correlation coefficient from 5° to 15°separation. Fig. S15 suggests that aggregations greater than 5° may over-smooth real variability.

We have included the new figure S9 in the supplement, and added the following text to the revised manuscript that reads "The 5°x5° and 5°x10° aggregation strike the necessary balance of reducing noise

(evidenced by the smoother IAV amplitude fields as aggregation increases in Fig. S1) but maintaining spatial information by not oversmoothing (evidenced by the fact that the aggregation occurs at spatial scales finer than the "elbow" where correlations among 1° gridcells stop changing with separation distance in Fig. S9.

[Figure]

**Supplementary Figure 8. Mean Correlation coefficient between the OCO-2 XCO₂ IAV of neighbouring gridcells in each 10° latitudinal band.**

*Reviewer's comments:*

*Sec 2.2.2: It seems like using a 3rd order polynomial on a 7-year time series to remove the secular increase is a recipe for problems when trying to derive the IAV. Wouldn't this artificially remove some of the IAV? Please discuss why 3rd order is necessary in the paper. Did you test 1st or 2nd order ? If so, why were they not sufficient?*

Authors' Response:

We use a 3rd order polynomial to derive the long-term trend for OCO-2, TCCON, MBL sites, and newly added GOSAT in the revision. When comparing the 1st and 2nd compared to 3rd, we didn't see a quantitative difference between the IAV patterns for OCO-2 (newly added Fig. S9). Although 1st or 2nd is sufficient for 7-years OCO-2, GOSAT observation is from 2009 to the end of 2021, and half of the TCCON sites we use have data more than a decade. Therefore, to guarantee our methods closely captures their trends and confirm that we are applying the same methodology to all the datasets included in our paper, we use the 3rd order polynomial fit. We added text in the methods section to justify the 3rd order fit that reads "We fit a third order polynomial to the raw timeseries since the GOSAT,MBL and TCCON

timeseries extend over a decade in length. We confirm that the use of a third-order polynomial, versus a second-order polynomial, does not remove the IAV signal from the shorter OCO-2 timeseries (Fig. S9)."

[Figure]

**Supplementary Figure 9. Timeseries comparison between the zonal mean GOSAT XCO₂ IAV, based on detrending method using 2ⁿᵈ and 3ʳᵈ polynomial fit.**

*Reviewer's comments:*

*Fig 5a: Care to comment on the strong feature near the beginning of 2020 peaking at 60°S latitude? That seems stronger that random variability.*

Authors' Response:

In section 3.1, we mentioned that "For the two El Ñino periods in 2015-2017 and late-2018 to 2021, high IAV values originate in the tropics and a smooth transition to high IAV values is seen at higher latitudes as time progresses (Fig. 5a)." The strong feature at 60°S near the beginning of 2020 looks suspicious, since when we explore the NOAA ESRL sites - SYO and PSA - around 60°S, there is no such pattern. This peaking based on OCO-2 does not look like random variability, and no potential related geophysical events can serve as good explanations , for example, no documented volcanic eruptions or fire events that could have caused it.   We explore if the anomaly peaking can persist after applying a more aggressive filtering - we additionally required that when calculating the average IAV of a certain zonal band of a certain month, there shall be observations for at least one third of the time - which is 24 out of 72 gridcells of available OCO-2 detection. With this aggressive filtering, the highly anomalous period was weakened yet still evident. We think the reasonable explanation of the peaking near the beginning of 2020 peaking at 60°S latitude is mainly due to OCO-2 issues with OCO-2 observation quality.

We have added the following text to the manuscript in Section 3.1, "While the OCO-2 patterns largely conform with variability expected based on ENSO and are in broad agreement with other observational networks, there are some anomalies that do not have an obvious explanation as of yet, such as the high $XCO_2$ in early 2020 around 60°S. Even with more aggressive data filtering, this episode persists, requiring more investigation of unknown geophysical drivers of high $XCO_2$ or potential retrieval issues that could cause a large positive bias."

[Figure]

**Figure 5. (a) Hovmöller Diagrams diagram showing zonal mean OCO-2 XCO₂ IAV timeseries for 5° latitude bins**

[Figure]

**IAV timeseries of NOAA ESRL sites around 60°S - SYO and PSA**

*Reviewer's comments:*

*Near Fig6: Because MEI/ENSO is such a heavily discussed topic in this work, a plot of the correlation coefficient of MEI with IAV timeseries in local 5x5 grid boxes may be warranted – similar to figure 6. Have you made such a plot, and does it show any interesting teleconnections? You may need to introduce a lag at the more northern latitudes when calculating correlation coefficients there (a simple 0-6 month lag as a function of latitude could work).*

Authors' Response:

We added a map (new Fig. 8) showing the correlation coefficient between the IAV timeseries with MEI, corresponding to Figure 4 which demonstrates the relationship between zonal-mean IAV and ENSO. With no month lag, we were able to see the mainly positive correlation over both ocean and continents, both Northern Hemisphere and Southern Hemisphere. We added text to section 3.1 of the paper to discuss this figure. The new text reads "We assess the spatial correlation patterns between the IAV

timeseries and MEI (Fig. 8a). The XCO$_2$ IAV timeseries have strong correlation coefficient with the MEI index in both Southern Hemisphere and Northern Hemisphere low latitudes from 0 to 30°N at lag 0, whereas in the Northern Hemisphere extratropic, the maximum positive correlation occurs at month 4 (Fig. 8b).  The positive correlation between MEI and the IAV timeseries is gradually attenuated, with no clear correlation at six months lag (Fig. 8c). ".

[Figure]

**Figure 8.  Correlation coefficient between local grid cell OCO-2 XCO$_2$ IAV timeseries and MEI .**

*Reviewer's comments:*

*Figure 10: Each "point" on the plot has in fact some uncertainty on the IAV at each site, due to both retrieval errors and spatiotemporal noise. Is it possible to get an estimate of this, and use it to add x & y error bars on each point? That might give a better picture of how consistent TCCON and OCO-2 IAV are, to within their respective errors. This figure implies that they are not very consistent. Related to the above, please check your IAV stddev calculations. I tried to reproduce your Bialystok numbers for OCO-2 and TCCON. Just by eyeballing your Fig S11, I got 1.05 for OCO-2 (similar to your number), but I got 0.75 for TCCON, whereas you got roughly 0.5. I wonder if some of your TCCON values are too low for some*

*reason (in particular at the NH sites). Your values at Karlsruhe and Orleans also both seem unreasonably low (both less than 0.4 ppm).*

Authors' Response:

We add x & y error bar for TCCON & OCO-2 on each point for the IAV based on the year-to-year difference of the seasonal cycle, and we see sites in the Northern Hemisphere are affected more by the errors and noises. We checked the IAV standard deviation calculation and the plotting of Figure 10, and found the mismatch between the TCCON IAV amplitudes and their locations, after the correction of location, we could see the accordance between this Figure 10 and supplementary S11 which shows the IAV timeseries.

[Figure]

**Figure 12. Comparison of OCO-2 and TCCON XCO₂ IAV amplitude at individual sites. Colours reflect site latitudes. The grey dashed line is the one-to-one identity line. The grey solid line is the error bar of the IAV amplitude.**

**Technical Comments**

*Reviewer's comments:*

*Line 72: Remove comma after "Chatterjee et al."*

Authors' Response:

We correct the wrong format and remove the comma after *"Chatterjee et al."*

*Reviewer's comments:*

*L76: "…is being used implicitly for flux attribution…". Please provide example references.*

Authors' Response:

We cite the reference: Nassar, R., Jones, D. B. A., Kulawik, S. S., Worden, J. R., Bowman, K. W., Andres, R. J., Suntharalingam, P., Chen, J. M., Brenninkmeijer, C. A. M., Schuck, T. J., Conway, T. J., & Worthy, D. E. (2011). Inverse modeling of $CO_2$ sources and sinks using satellite observations of $CO_2$ from TES and

surface flask measurements, *Atmospheric Chemistry and Physics*, 11, 6029–6047.
https://doi.org/10.5194/acp-11-6029-2011.

*Reviewer's comments:*

*L85: Please add reference Baker et al., Geosci Mod. Dev., https://doi.org/10.5194/gmd-15-649-2022.*

Authors' Response:

We add the reference *Baker et al., Geosci Mod. Dev., https://doi.org/10.5194/gmd-15-649-2022, for "This is especially important in light of analysis which suggests that the error variance budget in OCO-2 observations is large and contains substantial spatially coherent signal".*

*Reviewer's comments:*

*L110: Replace the O'Dell et al, 2012 reference with the O'Dell et al, 2018 reference. The former applies to GOSAT; the latter applies to OCO-2 and is a much more appropriate reference.*

Authors' Response:

Thanks for pointing out misusage of citation, we replace the O'Dell et al, 2012 reference with the O'Dell et al, 2018 reference for the sentence "After filtering and bias correction, the OCO-2 $XCO_2$ retrievals agree well with TCCON in nadir, glint, and target observation modes, and generally have absolute median differences less than 0.4 ppm and Root Mean Square differences less than 1.5 ppm."

*Reviewer's comments:*

*Sec 2.1.2: Please state somewhere if you use GGG2014 or GGG2020 (I'm assuming the former).*

Authors' Response:

We specify the usage of GGG2014 with the sentence, "Data are publicly available from the TCCON GGG2014 Data Archive (https://tccondata.org/) hosted by the California Institute of Technology."

*Reviewer's comments:*

*Sec 2.1.3: If there is any kind of version number or data source website for the NOAA sampling data, please provide it.*

Authors' Response:

We add the data source of NOAA MBL sampling data: "To explore differences in surface and column-average $CO_2$ IAV, we analyze IAV in the surface $CO_2$ mole fraction at  marine boundary layer (MBL) sites in the NOAA (National Oceanic and Atmospheric Administration) cooperative sampling network(https://gml.noaa.gov/dv/site/?program=ccgg)."

*Reviewer's comments:*

*Figure 1: Most of the TCCON sites are in completely wrong places!! It looks like the longitudes are screwed up?*

Authors' Response:

We express our thankfulness for the review pointing out the problem in mapping the TCCON locations.

We made corrections in the locations map Figure 1, confirming that TCCON sites in right places.

[Figure]

**Figure 1. Map showing the locations and the acronyms of the TCCON sites.**

*Reviewer's comments:*

*L161: Please provide a reference for the NOAA monthly OLR data set, or remove the sentence about the source of the ENSO-related variables (which is probably not necessary as it will be given in the MEI documentation).*

Authors' Response:

We remove the unnecessary sentence describing NOAA monthly OLR dataset.

*Reviewer's comments:*

*Figure 9 caption: Do you show R or R2 in panel b? Please make the caption more clear. Currently the caption says R and the plot says R2 on the axis label.*

Authors' Response:

We would like to show correlation coefficient R in Figure 9 panel b, the caption is now corrected.

[Figure]

**Figure 11. Latitudinal profile of regression Slope (panel a) and correlation coefficient (R, panel b) of OCO-2 versus TCCON XCO₂ IAV. The Slope and R values are based on using monthly XCO₂ IAV. The error bars result from a Monte Carlo bootstrapping approach . The colours represent the number of months data which used for the regression calculation, given gaps in both the OCO-2 and TCCON datasets.**

*Reviewer's comments:*

*Line 362: Change "R" to "correlation coefficients R". Otherwise, we really have to guess that you mean correlation coefficient.*

Authors' Response:

We clarify the misleading sentence and now it reads like "We derive the regression slopes and Correlation Coefficient R between OCO-2 and monthly averaged TCCON IAV through bootstrapping Linear Regression fitting techniques to investigate the coherence between IAV signals from space-based and in-situ ground-based observations."

*Reviewer's comments:*

Line 387: "although we note that the IAV amplitude is a factor of almost two smaller in the column average mole fraction". Relative to what? Please add "relative to boundary layer CO₂" or something similar.

Authors' Response:

We clarify the misleading sentence and now it reads like "All datasets show consistent patterns in the response to the El Ñino periods, although we note that the IAV amplitude is a factor of almost two smaller in the column average mole fraction compared to the boundary layer CO₂, which reflects the fact that IAV variations emerge due to surface fluxes in the lower part of the atmosphere,…"

*Reviewer's comments:*

*Fig S1 Caption: Please give the min # of soundings per gridbox.*

Authors' Response:

We change the caption of Fig S1 into "The IAV amplitude map, with different resolution from (a) 2.5° longitude by 2.5 ° latitude , to (b) 5 ° longitude by 5 ° latitude, to (c) 10 ° longitude by 10 ° latitude , to (d) 5 ° longitude by 10 ° latitude and (e) 5 ° longitude by 15 ° latitude, each gridbox has at least 5 soundings."

*Reviewer's comments:*

*Fig S3 Caption: Please give the spatial gridding (5x5, etc.) used, and change word "use" to "using" in the caption.*

Authors' Response:

We change the caption of Fig S3 into "The IAV amplitude map, using different sounding numbers as the benchmark to filter and get the aggregated 5°x5° OCO-2 detected $XCO_2$."

*Reviewer's comments:*

*Fig S6 Caption: Suggest changing word "record" to "years" ? It took me a while to figure out what you were getting at here.*

Authors' Response:

We change the caption of Fig S6 into "The number of valid records (X out of 6 for JAN~JULY 5 for AUG or 7 for SEP~DEC) for each month (JAN, FEB, etc...) for each 5°x5° gridcell."

---

## Author Comment (AC2)

**General comments:**

*Reviewer's comments:*

*Given that the IVA at MBL sites is almost double the IAV in the OCO-2 time series (Fig. 4a and 4b), Has the author tried to estimate XCO$_2$ at the lower layer of the troposphere? Some studies (e.g., Kulawik et al., (2017)) split the XCO$_2$ GOSAT column into two partial columns (lower <25 km elevation) and above 25 km., and found that the lower tropospheric CO$_2$ partial column compares well to the independent surface CO$_2$ data at oceanic sites. I wonder whether doing something similar here could improve the IAV comparison between OCO-2 and MLB.I also agree with reviewer 1, and a comparison with GOSAT XCO$_2$ satellite data will also be beneficial.*

Authors' Response:

We agree with the reviewer that it would be interesting to compare upper and lower partial column IAV with that of the MBL, but feel this is outside the scope of our analysis. We are not necessarily interested in perfect comparisons between OCO-2 IAV and that of the MBL, but rather showing that IAV signals from OCO-2 emerge from sources of error, given that IAV and error are of comparable magnitude for OCO-2 (Mitchell et al., 2023). We have added the following text to the discussion section: "In the future, as partial column retrievals (e.g., Kulawik et al., 2017) mature, intercomparisons of lowermost tropospheric partial columns may provide a useful bridge between variations in surface MBL observations and total column observations."

Both of the reviewers point out we could apply the method to GOSAT, with 4 years more data from 2009 to 2014. We added analysis based on GOSAT, which shows the similar patterns as OCO-2, and we are able to see that OCO-2 has the big improvements in terms of the data quality issue, as expected. We have added revised text to Section2.1.4 to describe the GOSAT dataset that reads "We compare patterns of XCO$_2$ IAV from OCO-2 with those from GOSAT. Also known as Ibuki, GOSAT is the world's first satellite dedicated to greenhouse gas monitoring, measuring global total column CO$_2$ and CH$_4$ since 2009. With the Thermal and Near infrared Sensor for carbon Observation (TANSO) - Fourier Transform Spectrometer (FTS) onboard for greenhouse gas monitoring using three SWIR bands and one TIR band (Cogan et al., 2012; Yoshida et al., 2013). Column-averaged dry mole fraction are obtained at a circular footprint of approximately 10.5 km. GOSAT has a regional biased of about approximately 0.3 ppm and 1.7 ppm single observation error versus the TCCON (Kulawik et al., 2016). We utilize the FTS SWIR Level 3 data global monthly 2.5° resolution mean CO$_2$ mixing ratio products from 2009 June to 2021 December to generate IAV and make comparisons with OCO-2. L3 products are generated by interpolating, extrapolating, and smoothing the FTS SWIR column-averaged mixing ratios of CO2 and apply the geostatistical calculation technique Kriging method. GOSAT observation datasets are available to public at NIES GOSAT website (https://www.gosat.nies.go.jp/en/about_5_products.html)."

We also added new Fig. 7 showing the GOSAT XCO$_2$ IAV amplitude, determined as the standard deviation of the IAV timeseries, with revised text in in section 3.2: "We carried out comparisons between the global spatiotemporal pattern of XCO$_2$ IAV between OCO-2 and GOSAT, since GOSAT has data beginning in 2009. The XCO$_2$ timeseries from OCO-2 provides higher coverage over mid-latitude oceans and tropical rainforests (stippling in Fig. 6, 7). The IAV amplitude of OCO-2 is generally smaller than that of GOSAT worldwide (Fig. 6, 7), which may be due to greater data volume and reduced noise in the OCO-2 dataset (Wu et al., 2020)"

**Specific comments:**

*Reviewer's comments:*

*Line 221: When averaged into broad zonal belts representing the tropics and mid-latitudes, the OCO-2 XCO₂ IAV time-series anomalies range between -0.5 to 0.75 ppm (Fig. 4). Does the author mean Fig.4a? Please indicates the results of Figs 4b and 4c are described in 3.2. For a moment, I thought that author was going to include the findings of MBL and TCCON in this section.*

Authors' Response:

Thanks for the opportunity to clarify the text. We specify that we are discussing Fig4.a, only referring to OCO-2 instead of MBL and TCCON. The revised text reads "When averaged into broad zonal belts representing the tropics and mid-latitudes, the OCO-2 XCO$_2$ IAV timeseries anomalies range between -0.5 to 0.75 ppm (Fig. 4a).".

*Reviewer's comments:*

*As an opinion, I believe that it would be better to only have one figure in this section (Fig 4a), and later in Section 3.2, I would include Fig 4a-4c as 4 different panels. For example, (a) (20-60S), (b) 0-20S, (c) 0N-20N and (d) 20-60N, where each panel should only contain the IAV time-series of OCO-2, MBL and TCCON together. By doing this, it would be easy for the reader to see how different the temporal-spatial variation is between these products.*

Authors' Response:

Thanks for offering this idea for the alternative way of timeseries comparison. We replotted the figure 4 and extended the time range of MBL/TCCON back to 2009 and remade the zonal mean timeseries following the reviewer's comments. The new version (as below) is still clear to see that MBL surface IAV is larger than TCCON/OCO-2/GOSAT, each type of observation is differentiable from another, and we can see the peak time changes from south to north.

With this new timeseries Fig.5 with GOSAT XCO$_2$ IAV included, we added the analysis in section 3.2, "OCO-2 and GOSAT zonal mean IAV timeseries generally share the same feature from 2014 to 2021, with an increasing trend during El Niño and decreasing trend during La Niña, however the GOSAT XCO$_2$ shows a delayed response in the northern midlatitudes, by almost 9 months, to the strong 2015 El Niño compared to the other datasets. Generally, GOSAT IAV timeseries are nosier, from month-to-month, compared to those from OCO-2. "

[Figure]

**Figure 4. IAV timeseries averaged for zonal bands between 60 °N and 60 °S from three different observing strategies. (a) Space-based OCO-2 XCO₂, (b) Surface CO2 observations from NOAA's marine boundary layer (MBL) sites, (c), Ground-based TCCON XCO₂, (d) Space-based GOSAT XCO₂. For all panels, the background shading indicates the Multivariate ENSO Index (MEI), which is positive during El Ñino phases.**

*Reviewer's comments:*

*Line 225: The Southern Hemisphere extratropical regions have larger and more rapid response in the IAV associated with ENSO compared to other zones, especially for a second. What does the author mean by saying: especially for a second?*

Authors' Response:

There is something problematic with the expression here, and we revise the sentence to make it clear we are talking about the time in the beginning of 2020. The revised text reads "The Southern Hemisphere extratropical regions have larger and more rapid response in the IAV associated with ENSO compared to other zones, especially for the smaller El Niño that peaked at the beginning of 2020. At this time, the $XCO_2$ IAV timeseries had an anomaly nearly twice as large as other latitude belts (Fig. 4)."

*Reviewer's comments:*

*The author also mentions that TCCON has a similar IAV amplitude to OCO-2. Is this true? Looking at Fig 4.c, TCCON has more IAV than OCO-2 (zonal belt (20-60S)). What happened in 2019? It seems that TCCON has an IAV of about 1.5 ppm compared to OCO-2 (no variability).*

Authors' Response:

We checked that there are only two sites: Lauder and Reunion Island in the 20-60°S belt. The spike is from the signal from Lauder. We didn't see this pattern from other nearby New Zealand NOAA ESRL sites, and we would suggest this spike could be due to limited TCCON sampling and data quality issue during wintertime and shouldering season, therefore we filtered out the IAV spike at zonal belt 20-60°S. The new timeseries is as below:

[Figure]

**IAV timeseries averaged for zonal bands between 60 °N and 60 °S based on Ground-based TCCON $XCO_2$**

*Reviewer's comments:*

*We note a slight low bias in OCO-2 relative to TCCON for all five sites in the Southern Hemisphere, which lie below the one-to-one line. How does the author know that OCO-2 has a lower bias than TCCON? I am a bit confused here; the author calculated standard deviations in Fig.10 and then discussed biases. Please clarify.*

Authors' Response:

The word 'bias' is used by mistake here. Just as the Reviewer points out, we are not talking about bias, but calculated the standard deviations of the timeseries for both OCO-2 and TCCON, so as to get the IAV amplitudes, and would like to the compare the amplitudes. The text now reads: 'We note a slight low

IAV amplitude in OCO-2 relative to TCCON for all five sites in the Southern Hemisphere which lie below the one-to-one line.'

*Reviewer's comments:*

*Line 427: When using IAV time series for flux inference, it will be crucial to account for non-flux imprints on the time series since spurious attribution of IAV will lead to biased fluxes. What does the author mean when he says: non-flux imprints on the time series? Flux or $XCO_2$? Spurious attribution of IAV in $XCO_2$ data?*

Authors' Response:

We give more explanation of the mentioned 'non-flux imprints on the timeseries,' which roughly refers to the imprints apart from fluxes, including atmospheric transport, random errors, systematic errors, and remote geophysical coherence. The revised text now reads "When using IAV timeseries for flux inference, it will be crucial to account for non-flux imprints such as imprint from atmospheric transport, random errors, systematic errors, and remote geophysical coherence on the timeseries (e.g., Torres et al., 2019; Mitchell et al., 2023), since spurious attribution of IAV will lead to biased fluxes."

**Editorial comments:**

*Reviewer's comments:*

*Table 1 and Table 2 should also be included in the appendix and not in the main text. Please be aware that TCCON locations must be placed correctly on the map. For example, reunion Island is over the Australian continent.*

Authors' Response:

We made corrections in the locations map Figure 1, confirming that TCCON sites in right places.

*Reviewer's comments:*

*Line 227: that of other latitude belts. Remove 'that of' from the text; it seems unnecessary in this sentence.*

Authors' Response:

We remove the redundant 'that of', now the sentence reads 'The Southern Hemisphere extratropical regions have larger and more rapid response in the IAV associated with ENSO compared to other zones, especially for the time of smaller El Niño at the beginning of 2020 when the MEI peaked, the $XCO_2$ IAV timeseries had an anomaly nearly twice as large as other latitude belts. During both El Niño events, the IAV timeseries in the NH tropics zone peaks nearly six months after the maximum MEI value. '

*Reviewer's comments:*

*El Niño instead of El Nino. Please be aware that El Niño is a Spanish word that must be written with 'Ñ'*

Authors' Response:

We went through the whole paper and changed all the miswritten *'El Nino'* into *'El Niño'.*

---

## Referee Report (RR1)

**Referee report: Manuscript titled "Temporal-Spatial Variations based on OCO-2 observations", by Yifan Guan et., al.**

I am satisfied with the author's responses to the questions raised from my initial review. I am pleased the author used GOSAT data as an independent assessment to provide robustness around findings found by OCO-2. Although the author added GOSAT in the results section, no discussion of GOSAT findings was added in the discussion or conclusion sections. It is worth adding this information given that both satellites showed agreement (Fig.4) in northern and southern hemisphere tropical zones (0-20°), with relatively small differences in Northern Hemisphere/Southern Hemisphere extratropical zones (20-60°) compared to TCCON and MBL.

I suggest the author re-write some parts of the abstract to convey main findings in the manuscript better. In line 40, the author says that 'similar zonal patterns of OCO-2 XCO2 IAV timeseries compared to ground-based in situ observations and with column observations from the Total Carbon Column Observing Network (TCCON) and the Greenhouse Gases Observing Satellite (GOSAT) provide validation that OCO-2 observations can be used reliably to estimate IAV'. Here the author uses the word "validation" when comparing OCO-2 and GOSAT. GOSAT is not used in this study to validate OCO-2 but to show robustness of the IVA of OCO-2. Here, I would say that assessment with independent satellite data, such as GOSAT, provides similar seasonal patterns with OCO-2 IVA and provide robustness to this study. Comparison with TCCON and MBL is more variable and suggests the largest variability of these estimates is related to poor sampling across zonal bands.

In the discussion, the author starts this section by saying: the good agreement among the OCO-2 XCO2 IAV timeseries, TCCON XCO2 IAV timeseries and the MBL surface CO2 IAV timeseries in broad zonal belts improves our confidence that we are able to quantify reasonable IAV timeseries from the satellite record. I can't entirely agree here with the author. Looking at Fig.4, I can clearly see that OCO-2 IVA is much smaller than TCCON and MBL surface data. I suggest to say that the OCO-2 show similar seasonal patterns to TCCON and MBL, but there is still a large amplitude difference between them. However, despite these differences, OCO-2 can still capture ENSO-driven variations and likely represent the IVA better due to the better spatial coverage compared to TCCON and MBL.

As mentioned at the beginning, there is no discussion about GOSAT here or in the conclusion section. Looking at Fig.4b and Fig4.c OCO-2 and GOSAT seems to agree well in the northern (20N-0) and southern hemisphere (0-20S), but with some differences in the northern (60N-20N) or southern extratropic (20S-60S). How well GOSAT compares to TCCON and MBL surface measurements at site level? I couldn't find this analysis in the result section or supplementary. Suppose GOSAT shows a poor agreement with TCCON in the extratropical bands. In that case, OCO-2 better characterize the IVA in these zonal bands compared to GOSAT and provides more accurate results to study IVA.

Minor editorial changes:

Add to Fig.4 the description of the zonal bands. For example, (a) northern hemisphere extratropical (60N-20N), (b) northern hemisphere (20N- 0), (c) Southern hemisphere (0-20S), (d) Southern hemisphere extratropical (20S-60S). I would also suggest to change the black colour of the OCO-2 time series to a more notorious colour.

Line 252. Which figure? I would write: the southern hemisphere extratropical region (Fig.1d). Remove the 's' at the end of regions. Here, I guess the author is referring to one zonal band.

Line 255. Why refer to Fig.4 only. If you are comparing Fig.4d to the other panels, you should write (fig.4a to c)

The caption of Figure 8 needs to be clarified. Is this figure showing lag correlations? If so, please indicate and add the description to label (a), (b) and (c) panels.

Line 257, We assess the spatial correlation patterns… should it says, we assess the spatial 'lag' correlation patterns.

Line 440, 452 : Again. El niño instead of El ñino. 'Ñ' goes in the second n. Check throughout the whole manuscript.

Line 273 the high XCO2 in early 2020 around 60°S. Which Figure?

Figure 10: (page 15) I suggest to indicate that the IAV timeseries come from XCO2 OCO-2.

Line 431. Fix typo in the word timeseries^are

Line 124 fix typo (1sigma) to 1-sigma

Please recheck the manuscript for typos and grammatical errors

---

## Author Response (AR2)

**Response to Reviewer**

**Reviewer's comments:**

*"Although the author added GOSAT in the results section, no discussion of GOSAT findings was added in the discussion or conclusion sections."*

**Author's Response:**

Thank you for pointing out this omission. We have now added to the discussion: "OCO-2 and GOSAT showed reasonable agreement (Fig.4) in northern and southern hemisphere tropical zones (0-20°), although there were some notable phase differences during the strong 2015 El Niño for GOSAT compared to the other timeseries in both the northern and southern extratropic regions. In contrast, OCO-2 shows good temporal agreement with the ground-based observations from MBL and TCCON."

**Reviewer's comments:**

*"I suggest the author re-write some parts of the abstract to convey the main findings in the manuscript better. In line 40, the author says that 'similar zonal patterns of OCO-2 $XCO_2$ IAV timeseries compared to ground-based in situ observations and with column observations from the Total Carbon Column Observing Network (TCCON) and the Greenhouse Gases Observing Satellite (GOSAT) provide validation that OCO-2 observations can be used reliably to estimate IAV'. Here the author uses the word "validation" when comparing OCO-2 and GOSAT."*

**Author's Response:**

We made changes to the Abstract: "Similar, but smoother, zonal patterns of OCO-2 $XCO_2$ IAV timeseries compared to ground-based in situ observations and with column observations from the Total Carbon Column Observing Network (TCCON) and the Greenhouse Gases Observing Satellite (GOSAT) show that OCO-2 observations can be used reliably to estimate IAV."

**Reviewer's comments:**

*"In the discussion, the author starts this section by saying: the good agreement among the OCO-2 $XCO_2$ IAV timeseries, TCCON $XCO_2$ IAV timeseries, and the MBL surface $CO_2$ IAV timeseries in broad zonal belts improves our confidence that we are able to quantify reasonable IAV timeseries from the satellite record. I can't entirely agree here with the author. Looking at Fig.4, I can clearly see that OCO-2 IAV is much smaller than TCCON and MBL surface data."*

**Author's Response:**

We rephrase the sentence to read: "The temporal agreement of the OCO-2 and TCCON $XCO_2$ IAV timeseries and the MBL surface $CO_2$ IAV timeseries in broad zonal belts improves our confidence that we can quantify IAV timeseries from the satellite record. We note that amplitude differences remain among the timeseries, owing to two major factors: first, compared to MBL surface observations, we expect $XCO_2$ timeseries to have smaller amplitudes of variability since it integrates over the entire atmospheric

column (Olsen and Randerson 2004), and second, the fact that the OCO-2 timeseries averages around a full latitude circle rather than a few discrete sites reduces some of the IAV contained in site-level records."

**Reviewer's comments:**

"As mentioned at the beginning, there is no discussion about GOSAT here or in the conclusion section. Looking at Fig.4b and Fig4.c OCO-2 and GOSAT seem to agree well in the northern (20N-0) and southern hemisphere (0-20S), but with some differences in the northern (60N-20N) or southern extratropic (20S-60S). How well GOSAT compares to TCCON and MBL surface measurements at site level? I couldn't find this analysis in the result section or supplementary. Suppose GOSAT shows a poor agreement with TCCON in the extratropical bands. In that case, OCO-2 better characterizes the IAV in these zonal bands compared to GOSAT and provides more accurate results to study IAV."

**Author's Response:**

Based on the reviewer's helpful comments above, we have added the following text to the discussion section, "OCO-2 and GOSAT showed reasonable agreement (Fig.4) in northern and southern hemisphere tropical zones (0-20°), although there were some notable phase differences during the strong 2015 El Niño for GOSAT compared to the other timeseries in both the northern and southern extratropic regions. In contrast, OCO-2 shows good temporal agreement with the ground-based observations from MBL and TCCON."

Our Fig. 7 shows that the IAV amplitude can be reliably calculated over a smaller fraction of the extratropic for GOSAT compared to OCO-2 (Figure 6), and we hypothesize that the lower number of observations and lower coverage leads to reduced agreement with GOSAT. We further note that the site-to-site TCCON and OCO-2 comparisons showed only modest agreements. Given that the focus of our paper is to understand IAV in the OCO-2 timeseries, not the GOSAT timeseries, and given the timeframe suggested by the editor for these revisions, we respectfully decline to perform site-level analyses with GOSAT or to state that the OCO-2 data are superior to GOSAT.

**Minor editorial changes**

**Reviewer's comments:**

*Add to Fig.4 the description of the zonal bands. For example, (a) northern hemisphere extratropical (60N-20N), (b) northern hemisphere (20N- 0), (c) Southern hemisphere (0-20S), (d) Southern hemisphere extratropical (20S-60S). I would also suggest to change the black color of the OCO-2 time series to a more notorious color.*

**Author's Response:**

[Figure]

We add the description for zonal bands - Caption becomes "IAV timeseries averaged for zonal bands between 60 °N and 60 °S from four different observing strategies: Space-based OCO-2 XCO$_2$ (Black), Surface CO$_2$ observations from NOAA's marine boundary layer (MBL) sites(Blue), Ground-based TCCON XCO$_2$ (Red), Space-based GOSAT XCO$_2$ (Gray). (a) temperate northern hemisphere (20°N-60°N), (b) tropical northern hemisphere (0° - 20°N), (c) tropical southern hemisphere (0°-20°S), (d) temperate

southern hemisphere  (20°S-60°S). For all panels, the background shading indicates the Multivariate ENSO Index (MEI), which is positive during El Niño phases."

We change the boldness of the Black lines for OCO-2 timeseries to make it clear to see and easy to compare with other observations.

**Reviewer's comments:**

*Line 252. Which figure? I would write: the southern hemisphere extratropical region (Fig.1d). Remove the 's' at the end of regions. Here, I guess the author is referring to one zonal band.*

**Author's Response:**

We corrected as the reviewer suggested: "The Southern Hemisphere extratropical region (Fig.1d)…"

**Reviewer's comments:**

*"Line 255. Why refer to Fig.4 only. If you are comparing Fig.4d to the other panels, you should write (fig.4a to c)."*

**Author's Response:**

We clarify that we are comparing the OCO-2 of Southern Hemisphere extratropical region shown in Fig. 4d to other regions shown in Fig. 4a to c. "At this time, the XCO$_2$ IAV timeseries(Fig. 4d) had an anomaly nearly twice as large as that of other latitude belts (Fig. 4a to 4c)."

**Reviewer's comments:**

*"The caption of Figure 8 needs to be clarified. Is this figure showing lag correlations? If so, please indicate and add the description to label (a), (b), and (c) panels."*

**Author's Response:**

We changed the captions to "Correlation coefficient between local grid cell OCO-2 XCO$_2$ IAV timeseries and MEI, for (a)synchronous timeseries,(b) with 3-month lags,(c) with 6-month lags."

**Reviewer's comments:**

*"Line 257, We assess the spatial correlation patterns… should it says, we assess the spatial 'lag' correlation patterns"*

**Author's Response:**

We change the sentence in to "we assess the spatial correlation patterns with no time lag, 3-month, 6-month lag".

**Reviewer's comments:**

*"Line 440, 452 : Again. El niño instead of El ñino. 'Ñ' goes in the second n. Check throughout the whole manuscript"*

**Author's Response:**

We checked through whole manuscripts and made the missed correction for "El Niño"

**Reviewer's comments:**

*"Line 273 the high XCO$_2$ in early 2020 around 60°S. Which Figure?"*

**Author's Response:**

We specify that we are looking at Fig. 5a when saying "...the high XCO$_2$ in early 2020 around 60°S..."

**Reviewer's comments:**

*"Figure 10: (page 15) I suggest to indicate that the IAV timeseries come from XCO$_2$ OCO-2"*

**Author's Response:**

We changed the captions to "Correlation coefficient between local grid cell IAV timeseries and the corresponding 5° zonal mean OCO-2 XCO$_2$ IAV timeseries".

**Reviewer's comments:**

*"Line 431. Fix the typo in the word timeseries^are"*

**Author's Response:**

We deleted the wrong typo '^'.

**Reviewer's comments:**

*"Line 124 fix typo (1sigma) to 1-sigma"*

**Author's Response:**

We corrected the typo from (1sigma) to 1-sigma